

# Sensitivity of NEMO4.0-SI[3] model parameters on sea ice budgets in the Southern Ocean

Yafei Nie[1,2,6], Chengkun Li[3], Martin Vancoppenolle[4], Bin Cheng[5], Fabio Boeira Dias[2], Xianqing Lv[6], Petteri Uotila[2]

[1]Frontier Science Center for Deep Ocean Multispheres and Earth System (FDOMES) and Physical Oceanography Laboratory, Ocean University of China, Qingdao, China
[2]Institute for Atmospheric and Earth System Research (INAR), Faculty of Science, University of Helsinki, Helsinki, Finland
[3]Department of Computer Science, University of Helsinki, Helsinki, Finland
[4]Laboratoire d'Océanographie et du Climat, CNRS/IRD/MNHN, Sorbonne Université, 75252, Paris, France
[5]Finnish Meteorological Institute, Helsinki, Finland
[6]Qingdao National Laboratory for Marine Science and Technology, Qingdao, China

*Correspondence to*: Petteri Uotila (petteri.uotila@helsinki.fi) and Xianqing Lv (xqinglv@ouc.edu.cn)

**Abstract.** The seasonally-dependent Antarctic sea ice concentration (SIC) budget is well-observed and synthesizes many important air-sea-ice interaction processes. However, it is rarely well simulated in Earth System Models and means to tune the former are not well understood. In this study, we investigate the sensitivity of 18 key NEMO4.0-SI[3] (Nucleus for European Modelling of the Ocean coupled with the Sea Ice modelling Integrated Initiative) model parameters on modelled SIC and sea ice volume (SIV) budgets in the Southern Ocean based on a total of 449 model runs and two global sensitivity analysis methods. We found the simulated SIC/SIV budgets are sensitivity to ice strength, the thermal conductivity of snow, the number of ice categories, two parameters related to lateral melting, ice-ocean drag coefficient and air-ice drag coefficient. A better quality of ice-ocean drag coefficient and air-ice drag coefficient can reduce the root-mean-square error between simulated and observed SIC budget by about 10%. We recommend ten combinations of NEMO4.0-SI[3] model parameters that could yield better sea ice extent, SIV seasonal cycles and SIC budgets than using the standard values.

## 1 Introduction

The Southern Ocean sea ice, a crucial component of the climate system, has experienced a slight but statistically significant expansion from 1979 to 2015 and remarkable fluctuations in the last few years (Comiso et al., 2017; Parkinson, 2019; Raphael and Handcock, 2022; Wang et al., 2022). Internal variability (Zunz et al., 2013; Mahlstein et al., 2013; Singh et al., 2019) and association with tropical oceans (Meehl et al., 2016; Li et al., 2021) have been used to understand the changes in sea ice, and there is consensus that atmospheric circulation, particularly wind, are primary drivers (Holland and Kwok, 2012; Matear et al., 2015; Hobbs et al., 2016). However, in contrast to observations, state-of-the-art climate models typically simulate a decline in Antarctic sea ice during this period (Zunz et al., 2013; Turner et al., 2013; Shu et al., 2015; Shu et al., 2020), the causes of which are subject to further diagnosis and identification.



Holland and Kwok (2012) proposed an analysis of sea ice concentration (SIC) budgets, i.e., decomposing the dynamic and the other processes leading to changes in SIC to compare with the same processes in observations, as an extension of the commonly used diagnostics for individual variables (e.g., SIC, ice thickness and ice drift). Diagnostics using SIC budgets for

fully coupled climate models as well as ocean-sea ice models driven by atmospheric reanalysis showed that the relatively realistic sea ice extent in the models was the result of excessive sea ice velocity bias (Uotila et al., 2014; Lecomte et al., 2016). Whereas correcting the sea ice velocity field in the model with satellite observations was able to simulate the trend of expanding sea ice extent in the Southern Ocean during 1992–2015 (Sun et al., 2021). Furthermore, correctly modelling the sea ice budget is so important as the ocean can only be driven correctly if the sea ice budget is realistic (Holmes et al., 2019),

which is related to the importance of sea ice in transporting fresh water (Abernathey et al., 2016; Haumann et al., 2016) and the role of sea ice as a mediator of polar air-ocean matter and energy exchange (Thomas and Dieckmann, 2010).

Sensitivity experiments with three different atmospheric reanalyses indicated that, at least in winter (April to October), SIC budgets are sensitive to atmospheric forcing, as sea ice models driven by these atmospheric reanalysis products show large errors compared to observations (Barthélemy et al., 2018). This was further validated by the fact that even when using

the same atmospheric reanalysis, the SIC budget in the ice-ocean reanalysis products can vary considerably (Nie et al., 2022). On the other hand, some studies have shown that simulations of the Southern Ocean sea ice area are not sensitive to model parameters (e.g., Massonnet et al., 2011; Uotila et al., 2012; Rae et al., 2014), but this is likely due to the dynamic and thermodynamic biases in SIC budget cancelling out (Uotila et al., 2014), i.e. wrong processes lead to a right-looking result. Therefore, a hypothesis was proposed that model physics could be more important than previously recognised for improving

sea ice modelling skills in the Southern Ocean (Barthélemy et al., 2018). Indeed, the conclusions of Uotila et al. (2014) showed that the SIC budget is sensitive to model configuration and they surmised that it may be possible to adjust the model parameters to make the SIC budget components more realistic. An example is that by changing the ice-ocean stress turning angle from 0° to 16°, the advection contribution to sea ice area change would be halved, although the divergence contribution would become unrealistic (Uotila et al., 2014). However, the sensitivity of the sea ice budgets to the model

parameters has not been systematically assessed to date.

The most common approach for sensitivity experiments is to adjust a single variable of interest at time, while keeping all other parameters fixed (e.g., Fichefet and Morales Maqueda, 1997; Rae et al., 2014), but due to the complexity and strong non-linearity of the model, there are often interactions between variables that cannot be identified with this approach. Another approach is to adjust several variables simultaneously. Kim et al. (2006) tested the sensitivity of 22 parameters of

the Los Alamos sea-ice model (CICE) based on the automatic differentiation method and adjusted the parameters to make the simulation as close as possible to the observations. Uotila et al. (2012) conducted experiments on 100 combinations of 10 parameters in a coupled ocean-ice model and recommended several optimal sets of parameters that would produce a realistic global sea ice distribution. To address the problem that the above sensitivity experiments cannot fully explore the entire high-dimensional parameter space, a more attractive way is to do a global sensitivity analysis (GSA; Saltelli et al., 2008).

However, a completely performed GSA requires a very large number of runs of the model, for example, $O(10^4)$ runs for





O(10) parameters (Saltelli et al., 2010). One option is to build an emulator to quickly and with modest computational requirements predict the possible model outputs for a given input and as a substitute for the full dynamic model (Sacks et al., 1989; Kennedy and O'Hagan, 2000; Oakley and O'Hagan, 2004). In brief, an emulator is a machine learning method that statistically constructs relationships between inputs and outputs from existing model results.

There has been some success in quantifying the parameter uncertainty using emulators in ocean/sea ice models. For example, Urrego-Blanco et al. (2016) applied a Gaussian process (GP) emulator to perform the GSA on 39 parameters in CICE. Williamson et al. (2017) built an emulator for the NEMO ocean model and quantified the effect of uncertainty on the model for 24 parameters. In this paper, our research objective is to quantify the sensitivity of the Southern Ocean SIC and sea ice volume (SIV) budgets to key parameters in a coupled ocean-sea ice model, and furthermore, to verify whether the

model parameters can be adjusted to obtain near-realistic SIC budget components. It is worth noting that NEMO4.0-SI[3] parameters' default values are generally optimised based on Arctic observations (e.g., Warren, 1999; Perovich et al., 2002; Lüpkes et al., 2012) and here we are investigating their optimal values in the Southern Ocean, which has not been done so far.

## 2 Materials and data

### 2.1 Model configuration and parameter space elicitation

Sea ice simulations in this study were performed using the version 4.0.7 revision 15731 of the Nucleus for European Modelling of the Ocean (NEMO; NEMO System Team, 2022) coupled with the Sea Ice modeling Integrated Initiative (SI[3]; NEMO Sea Ice Working Group, 2019), hereafter called NEMO4.0-SI[3]. The model represents global ocean via a commonly used nominal 2° tri-polar grid (ORCA2), which is about 85 km resolution between 55°S and 75°S. The ORCA2 was chosen

because it is already capable of identifying features of the Southern Ocean SIC budget at this resolution (Nie et al., 2022) and, considering that hundreds of experiments will be performed, using ORCA2 is computationally comparably cheap. The ORCA2 grid configuration has 31 unevenly spaced vertical layers from 10 m (near surface) to 500 m (at 5500 m depth). The vertical physics of the ocean is solved by the combination of the Turbulent Kinetic Energy (TKE) turbulent closure scheme (Marsaleix et al., 2008), an enhanced vertical diffusion scheme applied on tracer (Madec et al., 1998) and a double diffusive

mixing (Merryfield et al., 1999).

The sea ice momentum equation is calculated by using the adaptive elastic-viscous-plastic method (Kimmritz et al., 2016, 2017), which is formulated on a C-grid and improved the numerical efficiency of the modified EVP scheme. The default setting for the sea ice thickness category is 5, with 2 and 1 layers of ice and snow respectively. The thermodynamic component of NEMO4.0-SI[3] includes the 1D energy-conserving model (Bitz and Lipscomb, 1999) and a time-dependent

vertical salinity profile (Vancoppenolle et al., 2009). The sea ice model uses the same 1.5-hour time step as the ocean model.

In this study, the NEMO4.0-SI[3] model is forced with the DRAKKAR Forcing Set version 5.2 (DFS5.2, Dussin et al., 2016), based primarily on the ERA-Interim with some corrections (Dee et al., 2011) and covering the time period 1979–2017.



The DFS5.2 provides the atmospheric field required for the NCAR bulk formula (Large and Yeager, 2004) in NEMO4.0-SI[3],
which includes 2 m air temperature, 2 m specific humidity, 10 m zonal and meridional wind speeds, mean sea level pressure,
downward long-wave and short-wave radiation, and the total and solid precipitation rates. In these atmospheric fields, the
frequency of radiation and precipitation is 1 day and the other frequencies are 3 hours. The spatial resolution of DFS5.2 is
approximately 80 km, close to that of ORCA2 in the Southern Ocean. The continental discharge rates followed the
climatological dataset of Dai and Trenberth (2002). The simulations are initialized at rest via the temperature and salinity
fields from the World Ocean Atlas 2018 monthly climatology (WOA18; Garcia et al., 2019), run from January 1979 to
December 2017, with only the last decade of model output (2008-2017) being used for analysis.

Our selection principles for parameters and their values were first to target the three compartments at a stake (air, ocean
and ice) and their interactions, and second, to act on uncertain and important processes. Ultimately, we selected 18
parameters (Table 1) to investigate the sensitivity of the sea ice budget to their uncertainties. The lower and upper bounds of
the parameters were elicited according to the listed references and the uncertainty intervals were suitably extended to avoid
under-sampling in the marginal regions. The standard values of the parameters used for the control experiment (CTRL) are
the default values for NEMO4.0-SI[3].

**Table 1.** The 18 parameters investigated, including their realistic ranges taken from the listed references.

| Category | Symbol | Description and unit | Low | Standard | High | Reference |
|---|---|---|---|---|---|---|
| | rn_pstar | Ice strength parameter [N/m2] | 5.00E+03 | 2.00E+04 | 3.50E+04 | Massonnet et al. (2014) |
| | rhos | Snow density [kg/m3] | 130 | 330 | 530 | Massom et al. (2001) and Warren et al. (1999) |
| | rhoi | Ice density [kg/m3] | 880 | 917 | 940 | Timco and Frederking (1996) |
| | rn_cnd_s | Thermal conductivity of the snow [W/m/K] | 0.1 | 0.31 | 0.5 | Maykut and Untersteiner (1971) and Lecomte et al. (2013) |
| | rn_beta | Coefficient beta for lateral melting parameter | 0.2 | 1 | 1.8 | Lupkes et al. (2012) |
| Ice/snow | rn_dmin | Minimum floe diameter for lateral melting parameter [m] | 2 | 8 | 14 | Lupkes et al. (2012) |
| | rn_alb_sdry | Dry snow albdo | 0.85 | 0.85 | 0.87 | Perovich et al. (2002) and Brandt et al. (2005) |
| | rn_alb_smlt | Melting snow albdo | 0.72 | 0.75 | 0.82 | Perovich et al. (2002) and Brandt et al. (2005) |
| | rn_alb_idry | Dry ice albdo | 0.54 | 0.6 | 0.65 | Perovich et al. (2002) and Brandt et al. (2005) |
| | rn_alb_imlt | Melting ice albdo | 0.49 | 0.5 | 0.58 | Perovich et al. (2002) and Brandt et al. (2005) |
| | rn_sal_gd | Restoring ice salinity, gravity drainage [g/kg] | 4 | 5 | 7.5 | Nakawo and Sinha (1981) |
| | jpl | Number of ice categories | 1 | 5 | 30 | Massonnet et al. (2019) |
| | rn_avm0 | Eddy viscosity [m2/s] | 1.00E-05 | 1.20E-04 | 1.50E-04 | Williamson et al. (2017) |
| Ocean | rn_avt0 | Eddy diffusivity [m2/s] | 1.00E-06 | 1.20E-05 | 1.50E-05 | Williamson et al. (2017) |
| | rn_deds | Magnitude of the damping on salinity [mm/day] | -20 | -166.67 | -180 | NEMO System Team (2022) |
| | rn_ce | Magnitude of the mixed layer eddy | 0.04 | 0.06 | 0.1 | NEMO System Team (2022) |
| Coupling | rn_cio | Ice-ocean drag coefficient | 2.00E-03 | 5.00E-03 | 8.00E-03 | Massonnet et al. (2014) |
| | Cd_ice | Air-ice drag coefficient | 8.00E-04 | 1.40E-03 | 2.00E-03 | Massonnet et al. (2014) |






## 2.2 Experimental design

The experimental flow chart for achieving the sensitivity analysis is shown in Fig. 1. We start with the definition of the parameter space (see Table 1); the next steps are to sample from this parameter space and run the NEMO4.0-SI$^3$ model separately with the sampled set of parameters (the sampling method is described in the next paragraph). The model output

based on adequate sampling is then diagnosed by focusing mainly on three sets of metrics, including the area integral of simulated SIC/SIV budget components and the root-mean-square error (RMSE) between simulated and observed SIC budgets (RMSE$_{SICB}$). It is necessary to train a GP emulator (to be described in Section 2.3) for each metric to be evaluated based on NEMO4.0-SI$^3$ simulations, as both GSA methods, i.e., PAWN method (Pianosi and Wagener, 2015) and Sobol method (Sobol, 2001; described in Appendix A), would require a large number of model runs to comprehensively explore

the parameter space with a huge computational demand. Finally, once the key parameters have been identified, we will recommend some of the parameter sets that provide results close to the observations.

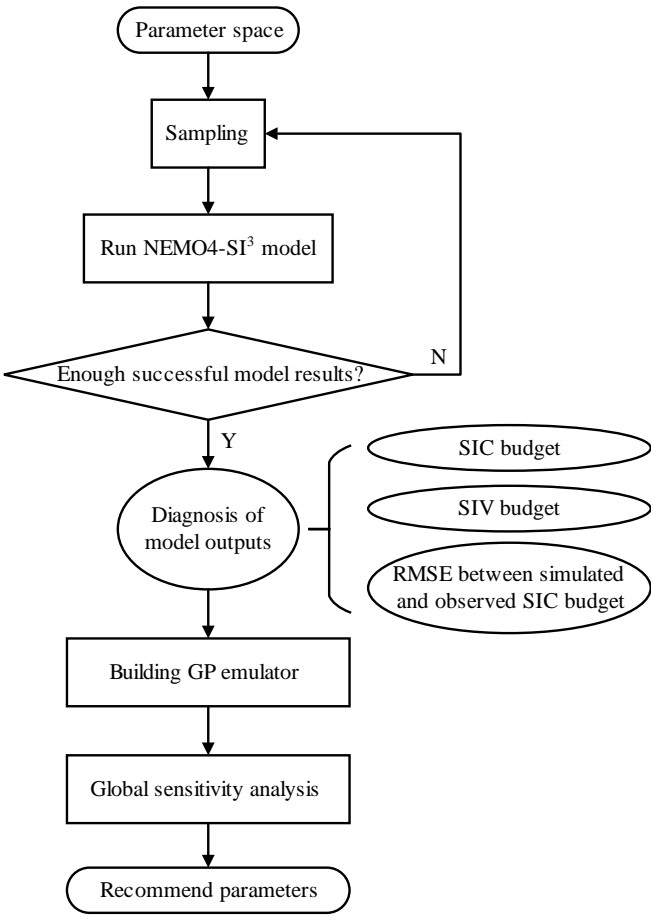

**Figure 1.** Experimental flow chart describing the sensitivity analysis.




We use the Latin Hypercube Sampling (LHS) method with a maxi-min property to generate low-discrepancy sequences from the 18-dimensional parameter space. The LHS is a stratified sampling method that divide each dimension evenly to ensure that samples are available in all intervals, and therefore allows for a more evenly drawn sample than the usual random sampling methods (Morris and Mitchell, 1995; McKay et al., 2000). Additionally, the maxi-min property is a space-filling

criteria that aims to maximize the minimum Euclidean distance between two sampling points and thus to improve the effectiveness of GP emulation (Joseph and Hung, 2008). The recommendation for the number of samples to build the emulator is N=10p (Loeppky et al., 2009), where p is the dimension of parameter space and equals to 18 in this study. In practice, however, we decided to use about 20p samples in order to build the GP emulator as accurate as possible (Williamson et al., 2017). Based on this principle, and taking into account possible model run failures, we first perform a

sampling of 800 points in parameter space to run the NEMO4.0-SI[3], and if the number of successful experiments ends up being too little (less than 360), we will continue the sampling.

## 2.3 Gaussian process emulator and model selection

In general, the PAWN method converges to a sufficiently accurate value with relatively fewer parameter adjustments than the Sobol method (Pianosi and Wagener, 2015), however, the amount of computation required for comprehensive model

evaluation remains too large, which requires the use of a GP emulator to make predictions about the output of the NEMO4.0-SI[3] after adjusting the parameters.

Let $X_t = (\vec{x}_1, \vec{x}_2, \cdots, \vec{x}_N)^T$ and $\vec{Y}_t = (y_1, y_2, \cdots, y_N)^T$ denote the total number of N simulations, each $\vec{x}_i$ is a p-dimensional column vector representing a sample of parameters, and each $y_i$ is a real number representing the corresponding model output, which is assumed to be noiseless here. A GP emulator $f(\cdot)$ of a model output variable $Y_t = f(X_t)$ can generally be

represented as

$$f(\cdot) \sim GP\big(\mu(\cdot), K(\cdot, \cdot)\big), \tag{1}$$

where $\mu(\cdot)$ and $K(\cdot, \cdot)$ are prior mean function and covariance function respectively. Then the posterior distribution for test parameter sets $X^*$ can be obtained as

$$f(\mathbf{X}^*)|f(\mathbf{X}_t) \sim N(\mu^*, K^*) \tag{2}$$

where

$$\mu^* = \mu + K(X^*, X_t)K(X_t, X_t)^{-1}(f(X_t) - \mu), \tag{3}$$

$$K^* = K(X^*, X^*) - K(X^*, X_t)K(X_t, X_t)^{-1}K(X_t, X^*). \tag{4}$$

We used the GPy implemented in Python (GPy, 2012) to build the GP emulator for each metric of interest. Once the user has selected the mean and covariance functions, the toolkit will automatically maximize the marginal likelihood by the L-

BFGS method to find the optimal values of all hyperparameters in the mean and covariance functions. In general, the prior of the mean function is assumed to be zero. Thus, the only key matter remaining is how to choose the covariance function.





To achieve this, we used a 10-fold cross-validation method for model selection (Geisser, 1975). The idea is to divide the dataset $\{X_t, Y\}$ evenly into 10 parts, each time using 9 parts as the "training data" to train the emulator and 1 part as the "true data" for model validation, and so on for 10 cycles and taking the average as a proxy for model performance. Using this approach, we traversed the linear, squared exponential, exponential, Matern 3/2, Matern 5/2 covariance functions and their sums and products (Rasmussen and Williams, 2006), for a total of 177 different combinations, and then selected the covariance function with both the minimum RMSE and the highest correlation coefficient between the simulated and emulated values.

### 2.4 Sea ice concentration and volume budgets

Following the ice conservation law, the change of a sea ice state field $\Theta$, such as SIC and SIV, can be contributed to dynamic and other processes (Leppäranta 2011, Chapter 3.4):

$$\frac{\partial \Theta}{\partial t} = -\boldsymbol{u} \cdot \nabla\Theta - \Theta\nabla \cdot \boldsymbol{u} + (f - r) \tag{5}$$

where $\boldsymbol{u}$ is the sea ice velocity, $f$ represents the change from freezing/melting, $r$ stands for any other progresses (e.g., ridging and rafting). Integrating the Eq.(5) in time, then the net changes in $\Theta$ over a period of time $(t_2 - t_1)$ can be obtained as:

$$\int_{t_1}^{t_2}\frac{\partial \Theta}{\partial t}dt = -\int_{t_1}^{t_2}\boldsymbol{u} \cdot \nabla\Theta dt - \int_{t_1}^{t_2}\Theta\nabla \cdot \boldsymbol{u}dt + \int_{t_1}^{t_2}(f - r)dt, \tag{6}$$

where the left-hand-side term is the change or $dadt$, the first term on the right-hand-side represents the contribution of advection ($adv$), the second term divergence ($div$) and the last term residual ($res$). A positive value for each term is defined as an increase of $\Theta$ and a negative value for a decrease.

The budgets for SIC and SIV were calculated in our study, including seasonal climatologies for each SIC or SIV budget term, following the same approach as Holland and Kimura (2016). First, the daily $dadt$ was obtained by central differencing of the ice fields on the day before and after; the advection and divergence were first calculated on each day, and then averaged over the corresponding 3-day periods to be consistent with the daily $dadt$. Second, $adv$ and $div$ were subtracted from the $dadt$ to obtain the daily $res$; and finally, all daily terms were summed over each season and averaged over the years 2008-2017.

### 2.5 Observation data

Daily sea ice velocity observations from Kimura et al. (2013) and SIC from the NOAA/NSIDC Climate Data Record of Passive Microwave Sea Ice Concentration, Version 4 (Meier et al., 2021) (hereafter referred to as CDR) were used to calculate the observed SIC budget. The ice velocity dataset KIMURA was generated from the brightness temperature of the 36-GHz channel of the Advanced Microwave Scanning Radiometer-Earth Observing System (AMSR-E) using the maximum cross correlation technique (Kimura et al., 2013), and ultimately deriving a 60 km resolution product. Therefore, the KIMURA data shares the same period as AMSR-E and its successor AMSR2, covering from 2002 to the present. Following



Holland and Kwok (2012), a $3 \times 3$ grid filter was used in the calculations to smooth out the grid-scale noise present in the
satellite-derived ice drift. Regarding the SIC satellite observations, the CDR SIC is a rule-based combination of the NASA
Team (Cavalieri et al., 1984) and NASA Bootstrap (Comiso, 1986) ice concentration datasets in the same $25\ km \times 25\ km$
grid, covering the years from 1978 to 2021, with daily, grid-based uncertainty estimates.

The observed SIC budget (Fig. B1) shows that the Southern Ocean sea ice is generally transported to the ice edge at lower
latitudes by advection and melts there, with divergence yields open water and thus promotes freezing of ice (Holland and
Kwok, 2012; Uotila et al., 2014). It is important to note that the calculated SIC budget observations were considered as "true
values" in our study, despite the uncertainties and biases in the ice drift observations, such as the overall overestimation of 5%
compared to the buoy measured velocities (Kimura et al., 2013). The simulated SIC budgets and the root-mean-square errors
from the observed one were only calculated at grid points with SIC larger than 15% and at dates where ice drift observations
existed, to minimize the uncertainty of results caused by missing observations and observational errors.

## 3 Results

### 3.1 Sea ice concentration and thickness in the model ensemble

Out of 800 experiments, 44% were terminated due to model instability caused by parameter combinations, resulting in an
ensemble of models of size 449, which included the CTRL experiment. The seasonal cycles of sea ice extent (SIE; integral
of grid cells areas where SIC > 15%) and area (SIA; integral of grid cells areas multiplied by the SIC in each grid cell) for
the model ensemble are shown in Fig. 2. The SIE and SIA intervals for the ensemble cover the observed values fairly well,
except for September when SIA is systematically slightly overestimated. Inter-model disagreement due to parameter
uncertainty is greatest in summer (ranging from 0.42 to $8.26 \times 10^6\ km^2$), when SIE and SIA are at a minimum (observed at
$4.26 \times 10^6\ km^2$), while there is little disagreement between models during the autumn months. Among the members of the
model ensemble, the CTRL run essentially overlaps with the ensemble mean and matches well with the observation.

In February, comparing the ensemble mean SIC (Fig. B2a-b) with the CDR observation shows that there are still
challenges in the modelling of the local patterns, especially as the NEMO4.0-SI[3] significantly underestimates the SIC near
the East Antarctic coast. In addition, the ensemble standard deviation for February stands at a high level (around 20%) in
most regions. Whereas in September (Fig. B2d-f) the ensemble mean SIC is more consistent with the observations than in
February, although differences between the ensemble members remain relatively high (around 10%) in marginal ice areas
where the SIC is low. Overall, the discrepancies between ensemble members due to parameter uncertainty are smaller at high
SIC areas (SIC > 90%) than in low SIC areas.





**Figure 2.** Simulated monthly climatologies of (a) sea ice extent (SIE), (b) area (SIA) and (c) volume (SIV) from 2008 to 2017, ensemble model means and results from four sets of experiments of interest are also highlighted. The SIE and SIA calculated from the NOAA/NSIDC Climate Data Record of Passive Microwave Sea Ice Concentration, Version 4 (CDR) are used as references.





Similar to the seasonal cycles of SIE and SIA, the CTRL run's SIV remains close to the ensemble mean. However, the differences between SIVs simulated based on different parameter sets are much greater than for SIEs (Fig. 2c), for instance

in winter, the maximum values of SIVs in the ensemble members are more than twice as large as the minimum values. Additionally, there are some model runs whose SIV cycles are detached from other members, which is most evident in winter. For the ensemble mean sea ice thickness, thicker sea ice of up to two meters is maintained year-round in the western Weddell Sea (Fig. B3a,c), which appears to be higher than the previous observation-based dataset of 1.2 to 1.5 meters (Haumann et al., 2016, in their Extended Data Figure 2). However, the lack of observations from the same period as this

study precludes a direct comparison. The spatial pattern of ice thickness standard deviation between model ensembles (Fig. B3) is similar to that of sea ice thickness, which means thicker sea ice is usually accompanied by a larger standard deviation.

Diagnostics of the SIC and sea ice thickness of the model ensemble show that the NEMO4.0-SI$^3$ model driven by DFS5.2 provides reasonable results. The mean states of the model ensemble being close to the CTRL experiment, for SIC in particular, match the observations very well, which provides a good basis for the following analysis of the budgets.

**3.2 Budgets on ice concentration and volume**

By applying the same approach as for the calculation of the observed SIC budget (cf. Fig. B1), in this section we calculate the SIC budget and SIV budget for the ensemble of 449 model runs. As can be seen in Fig. 3, the spatial pattern characteristics of the ensemble mean of $dadt$ and $adv$ for each season are generally consistent with observations. The magnitudes of the model ensembles of $dadt$ and $adv$ are significantly larger due to the fact that the observed ice drift has

some missing values and the $dadt$ term is only integrated over the grids with ice drift observations. However, the simulated divergence appears to be systematically biased when compared to the observation, the simulated $div$ in the inner ice pack is smaller than the observed even considering there are missing data in the observation, and some sporadic convergence (positive value of divergence) scattered in the marginal ice zone is not captured by the model. The lack of divergence in the inner ice pack also leads to a lack of open water and thus insufficient freezing of sea ice, which can be seen from the winter

and spring $res$ in Fig. 3, and in summer in the south Weddell Sea. In summer, the overall contribution of model simulated advection and divergence to sea ice change is minimal, with thermodynamic sea ice melt dominating, which is consistent with the observation.

The standard deviation of each budget terms for the model ensemble was also calculated (Fig. B4), the deviations between simulated sea ice changes are mainly concentrated in autumn and summer, and are mainly located in the Weddell and Ross

Seas, with insignificant deviations in winter and autumn. For the advection term, the inter-model deviation is large at the ice edge, where sea ice is transported by the advection, and the coastal area, where winds and currents are strong. The deviations of the divergence term in the model ensemble are mostly concentrated in the coastal region, while the model ensemble is more consistent in the inner ice pack, although the greatest differences between simulations and observations are found there. Since the $res$ term was calculated by subtracting $adv$ and $div$ from $dadt$, the deviations in these three terms are generally




combined in the *res* term, with the possible exception of some cancelling out of deviations in these terms, for example, in the Weddell Sea in autumn *res* deviates less than *dadt*.

**Figure 3.** Mean seasonal SIC budget components for the ensemble of 449 model runs from 2008 to 2017. The SIC budget
for each member was first calculated separately and then averaged together.

The SIV increases extensively in the Southern Ocean in autumn and winter and decreases in summer (*dadt* column in Fig. 4), and is generally decreasing in spring, except for a slight increase in the Amundsen-Bellingshausen Seas as well as along the South Weddell Sea. Differing from the SIC budget (Fig. 3) in which advection contributes little to sea ice changes in the





inner ice pack, the ensemble model mean shows that advection will lead to a reduction in SIV ($adv$ column in Fig. 4), although SIC maintains high in this region. The spatial pattern of the divergence of SIV is very little different from that of SIC, and since the contribution of simulated SIC divergence to sea ice change is underestimated compared to the observation as mentioned earlier, it is safe to assume here that divergence should similarly underestimate the change in SIV, given the strong interdependence of SIC and SIV. The inner ice pack maintains an increase in SIV from autumn to spring as the sea ice

freezes, and from spring onwards the sea ice starts to melt from the marginal ice zone and reaches a full melting of the entire Southern Ocean sea ice in summer ($res$ column in Fig. 4).

**Figure 4.** As Fig. 3, but for SIV budget.






For simulations of overall changes in SIV, the standard deviation between ensemble members is only slightly greater in summer than in other seasons (Fig. B5). The disagreement between members originates mainly from the contribution of advection to SIV change, which is most pronounced along the West Weddell Sea and Antarctic Peninsula coasts, in marginal ice zone and the East Antarctic coast. In addition, the contribution of advection and divergence to SIV that simulated based

on different parameter sets, varies considerably in the Antarctica coastal region, similar to the SIC budget. The residual term still has the largest standard deviation as it retains the deviations of the other terms.

The area integrals of each budget term for the simulated SIC and SIV are presented in Table 2. Although this quantification of the contribution of each term to sea ice change does not consider local differences and cancels out positive and negative sea ice change to some extent, it is a simple and easy to implement a method for quantifying the sensitivity of

sea ice budget to parameters. As can be seen from the ensemble mean of SIC and SIV budget terms, the area integrals of the advection and divergence contributions to sea ice change largely cancel each other out, which is potentially because these two processes do not change the total amount of sea ice. This also means that when studying the effect of model parameter uncertainty on sea ice budget in the following sections, it is only necessary to use the area integrals of $res$ (or $dadt$) and $adv$ (or $div$).


**Table 2.** Area integrals of sea ice concentration (SIC) and sea ice volume (SIV) budget components for the ensemble of 449 model runs. Data are listed in the form of mean $\pm$ one standard deviation. The units are $10^6 \ km^2$ and $10^3 \ km^3$ for SIC and SIV budget respectively.

| Season | Name | dadt | adv | div | res |
|---|---|---|---|---|---|
| Autumn (MAM) | SIC | $8.57 \pm 0.47$ | $2.30 \pm 0.22$ | $-2.35 \pm 0.22$ | $8.62 \pm 0.47$ |
| | SIV | $9.51 \pm 1.06$ | $2.23 \pm 0.42$ | $-2.17 \pm 0.41$ | $9.45 \pm 1.05$ |
| Winter (JJA) | SIC | $6.74 \pm 0.17$ | $3.17 \pm 0.37$ | $-3.28 \pm 0.38$ | $6.85 \pm 0.18$ |
| | SIV | $18.73 \pm 2.13$ | $4.94 \pm 0.87$ | $-4.75 \pm 0.86$ | $18.55 \pm 2.11$ |
| Spring (SON) | SIC | $-5.84 \pm 0.73$ | $2.91 \pm 0.35$ | $-3.02 \pm 0.35$ | $-5.73 \pm 0.72$ |
| | SIV | $-5.86 \pm 2.01$ | $6.27 \pm 1.05$ | $-6.02 \pm 1.04$ | $-6.10 \pm 2.04$ |
| Summer (DJF) | SIC | $-9.57 \pm 0.40$ | $0.55 \pm 0.11$ | $-0.55 \pm 0.11$ | $-9.57 \pm 0.40$ |
| | SIV | $-22.65 \pm 3.01$ | $1.02 \pm 0.29$ | $-1.00 \pm 0.29$ | $-22.67 \pm 3.01$ |


The RMSE$_{SICB}$ is calculated as a complement to the area integrals of each SIC budget term. In matching the simulated results to the observation, we first linearly interpolated the modelled data onto the grid cells containing observed data, and then calculated daily budgets for only those dates for which observations were available and for grids with SIC greater than 15%, and finally calculated the seasonal SIC budget climatology. Fig. 5 counts the RMSE$_{SICB}$ for all model ensemble

members. The model ensemble has the smallest RMSE$_{SICB}$ with observations in term of sea ice change (~15%), followed by



advection (~25%), and a larger $RMSE_{SICB}$ for the divergence term, which is consistent with the results showed in Fig. 3 and Fig. B1. In the model ensemble, the $RMSE_{SICB}$ of the CTRL experiment is essentially at or below the median level, and the distributions of the $RMSE_{SICB}$ in the model ensemble are not symmetric, i.e., there are more flier points outside of third quartile plus 1.5 times the inter-quartile range.

Based on the results of this section, the area integrals of $adv$ and $res$ in the SIC (and SIV) budget and the $RMSE_{SICB}$ are used as the metrics to assess the sensitivity of the model's sea ice budget to 18 parameters in the next sections.

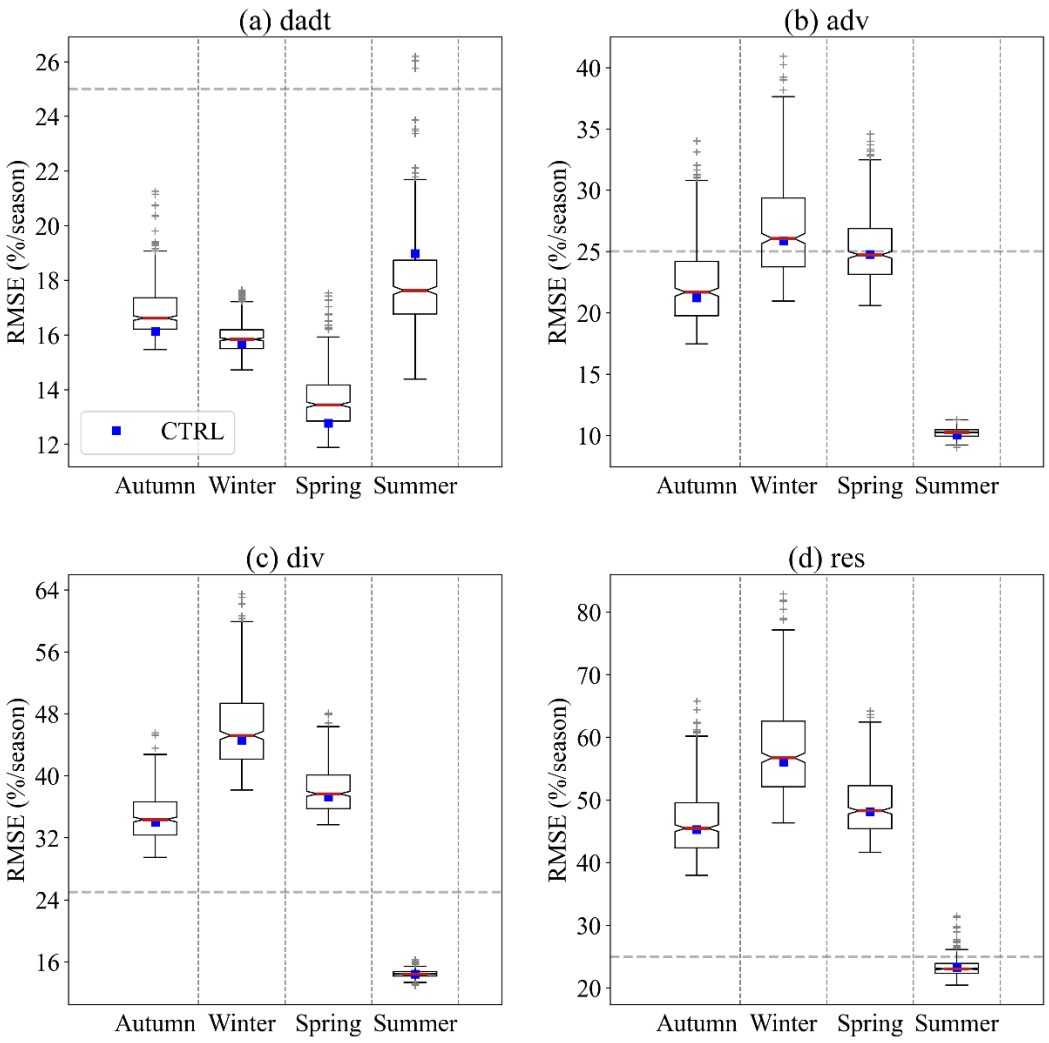

**Figure 5.** Boxplots of RMSE for each component of the simulated and observed SIC budget. Boxes extend from the first
quartile (top border) to the third quartile (bottom border), the red line represents the median of all 449 model results and the blue squares represent the CTRL experiment. The whiskers extend outwards from the box to 1.5 times the inter-quartile range, with a few flier points beyond the whiskers. The 25% horizontal dashed lines are marked as references.





### 3.3 Sensitivity of ice concentration and volume budgets to parameters

Before conducting the GSA, Fig. B6 shows the cross-validation results for the best GP emulator for each of the $adv$ and $res$
term area integral metrics of the SIC and SIV budgets. Overall, the emulated and simulated values have a very high
correlation coefficients (typically greater than 0.98), thus the built emulator is considered successful and will be used as a
proxy for NEMO4.0-SI[3] in the subsequent sensitivity analysis.

The sensitivity of each metric to the 18 parameters, quantified by the Sobol and PAWN methods, is illustrated in Fig. 6. It
should be noted that the sensitivity scores for the two methods are independent and not comparable in absolute terms.
Following Urrego-Blanco et al. (2016), the Sobol sensitivity index below 0.02 is considered insignificant, and for the
Kolmogorov-Smirnov (KS) mean index in PAWN, the critical value at confidence level of 0.05 is about $6.65 \times 10^{-2}$. Both
GSA methods show that the advection is very sensitive to ice strength (rn_pstar) outside of summer in the SIC budget. Ice-
ocean drag coefficient (rn_cio) and air-ice drag coefficient (Cd_ice) have an influence on the modelled advection
contribution to sea ice change from summer to autumn and spring, respectively. In summer, the snow thermal conductivity
(rn_cnd_s) and two lateral melting parameters (rn_beta and rn_dmin) also has some effect on the advection of SIC budget.
The total and first-order Sobol indices are not very different, which is usually the case for both indices of the PWAN method,
however, with the exception of the jpl (ice category number), where KS max is shown to be much larger than KS mean (e.g.,
in autumn and summer). For other metrics, this also happens for sensitivity assessment of some other parameters, which will
be discussed further in the next section. The residual term of the SIC budget shows considerable sensitivity to rn_cio, which
persists from autumn to spring. Meanwhile the effect of Cd_ice on $res$ increases continuously from autumn to summer. Ice
strength still has a weak effect, much less than its effect on $adv$. In addition, rn_cnd_s and jpl have a non-negligible effect
on the modelling of $res$ in winter and summer, respectively.

Among the sensitivity indices of the SIV budget, the most noticeable parameter is rn_cnd_s, to which both $adv$ and $res$
are very sensitive at all times of the year, except in the spring when it has less impact on $res$ (Fig. 6). Another physical
parameter related to the snow on sea ice (rhos, i.e., snow density) is important for $res$ simulations in the SIV budget,
especially from autumn to winter, the period when sea ice freezes fast (Fig. 2c). Similar to the SIC budget, the rn_cio and
Cd_ice remain crucial for the SIV budget in spring and summer, while the ice strength is only important for advection in
winter and spring.






**Figure 6.** The total (ST) and first-order (S1) Sobol sensitivity indices, and the maximum (KS max) and mean (KS mean) PAWN sensitivity indices for each sea ice budget component to 18 parameters. The blue and grey dashed lines are the thresholds for S1 and KS mean indices, respectively. Larger Sobol/PAWN index value indicates that the metric is more sensitive to this parameter. The blue connecting line indicates that the Sobol second-order index for the combination of these

two parameters is greater than 0.02.





### 3.4 Sensitivity of SIC budget errors to parameters

The results for four $RMSE_{SICB}$ metrics based on the best performing GP emulators are shown in Fig. B7. The GP emulator performs perfectly for the $RMSE_{SICB}$ of $adv$, $div$ and $res$, with a correlation coefficient greater than 0.998, except in summer. As can be seen in Fig. 5 in the summer months, the difference in $RMSE_{SICB}$ for these three terms is very small

compared to other seasons, and this small difference is likely to be random and therefore difficult to capture well by the GP emulator. The GP emulator also does not perform well in terms of $dadt$ $RMSE_{SICB}$ (Fig. B7, first column) and there is also likely to be a large randomness in the difference in $dadt$ between the model ensemble and the observation. Given the poor performance of the GP emulator in terms of $dadt$ $RMSE_{SICB}$ as well as $RMSE_{SICB}$ over the summer, the GSA results obtained by using it instead of the NEMO4.0-SI[3] dynamical ocean model are subject to uncertainty and should be kept in

mind in the following analysis.

Fig. 7 demonstrates quite clearly that for $adv$, $div$ and $res$ $RMSE_{SICB}$ in autumn, winter and spring (which are also the terms and seasons with the largest $RMSE_{SICB}$ values, Fig. 5), only air-ice and ice-ocean drag coefficients are the most critical parameters, while ice strength also has, but only weakly, an effect. Besides these two important drag coefficients, Fig. 7 also shows that the $dadt$ $RMSE_{SICB}$ between model and observation might be sensitive to the snow thermal conductivity and ice

category number to some extent. The analysis is more complicated in summer, as is the sensitivity of SIC budget and SIV budget to the parameters. In addition to all the previously mentioned parameters that have an impact, Fig. 7 shows that in summer the $RMSE_{SICB}$ may also be sensitive to the minimum floe diameter for lateral melting parameter (rn_dmin) and the magnitude of the damping on salinity (rn_deds), which is a parameter belonging to the ocean module. Further comparing Fig. 6 and Fig. 7, it can be found that overall, both the simulation of the SIC budget by the NEMO4.0-SI[3] model and its

$RMSE_{SICB}$ are most sensitive to the air-ice and ice-ocean drag coefficients, both of which belong to the coupling category in Table 1. Next important are ice strength as well as the thermal conductivity of snow, identified by the six metrics related to SIC budget. In summer, some thermodynamic melting related parameters, such as rn_beta and rn_dmin, are important. In contrast, the SIC budget simulated by the model is sensitive to jpl, unlike the $RMSE_{SICB}$ metrics.

As it has been identified that the $RMSE_{SICB}$ metrics are sensitive to the two most critical parameters (rn_cio and Cd_ice)

and one relatively important parameter (rn_pstar), Fig. 8 illustrates the $RMSE_{SICB}$ for all SIC budget terms and all seasons, averaged over 449 model runs, in relation to the values of these three parameters with the top 10 combinations listed in Table 3. It can be seen in Fig. 8b that the $RMSE_{SICB}$ broadly decreases with increasing rn_cio and decreasing Cd_ice, such that the 10 sets of model runs with the smallest $RMSE_{SICB}$ are concentrated in the top left corner of the figure, where Cd_ice is approximately from $8 \times 10^{-4}$ to $1 \times 10^{-3}$, and rn_cio is approximately from $5.5 \times 10^{-3}$ to $7.5 \times 10^{-3}$ (Table 3). In

contrast, the best 10 ice strength values are more dispersed, and greater than $15 \times 10^{3}$ (Fig. 8a,c), and the $RMSE_{SICB}$ does not depend linearly on it as with Cd_ice and rn_cio.

**Figure 7.** As Fig. 6, but for the sensitivity of the RMSE between SIC budgets of the model and the observation to 18 parameters. The red connecting lines are the same as the blue ones but for the Sobol second-order index larger than 0.1.



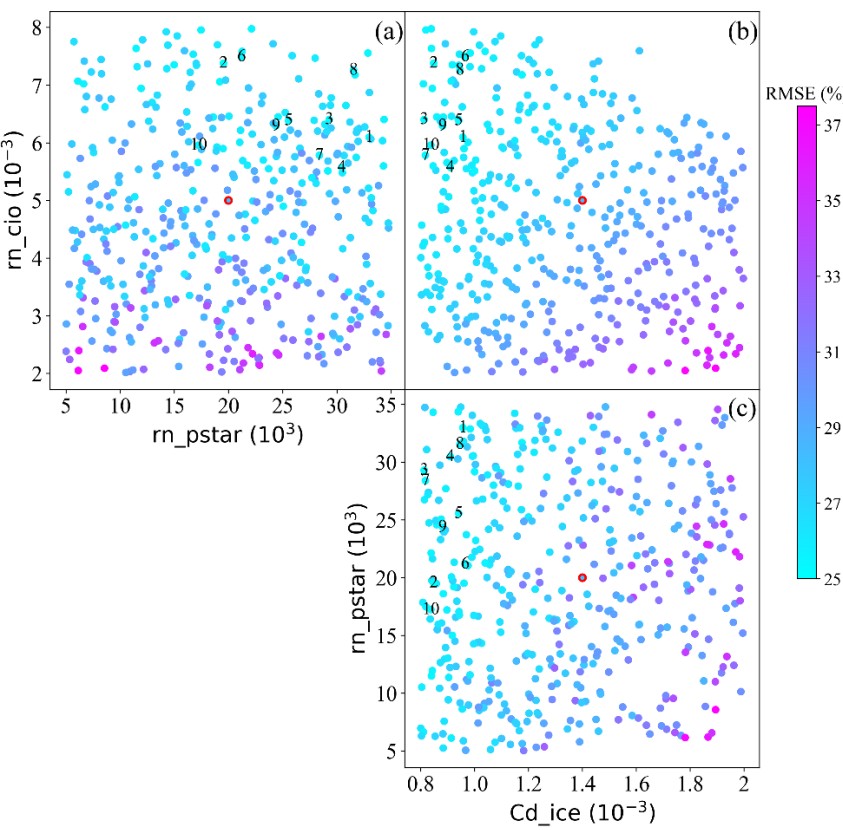

**Figure 8.** Average RMSE$_{SICB}$ for all four SIC budget components for different combinations of key parameters. The numbers 1 to 10 indicate the results of the 10 best parameter sets in ascending order of the average RMSE$_{SICB}$, and the points with red edges indicate the standard values used for the CTRL experiment.

**Table 3.** The 10 best performed experiments in terms of mean RMSE$_{SICB}$ (i.e., RMSE between simulated and observed SIC budget) and the values of the 3 key parameters they used. Note that these values are highly correspond to the DRAKKAR Forcing Set version 5.2 (Dussin et al., 2016) atmospheric forcing used in this study.

| Rank | RMSE (%) | Cd_ice ($10^{-4}$) | rn_cio ($10^{-3}$) | rn_pstar ($10^4$) |
|------|----------|-----------|----------|-----------|
| 1 | 25.127 | 9.563 | 6.094 | 3.298 |
| 2 | 25.163 | 8.478 | 7.379 | 1.954 |
| 3 | 25.182 | 8.125 | 6.402 | 2.929 |
| 4 | 25.270 | 9.100 | 5.572 | 3.047 |
| 5 | 25.299 | 9.407 | 6.384 | 2.555 |
| 6 | 25.356 | 9.643 | 7.491 | 2.119 |
| 7 | 25.364 | 8.172 | 5.783 | 2.839 |
| 8 | 25.378 | 9.455 | 7.262 | 3.154 |
| 9 | 25.389 | 8.807 | 6.293 | 2.437 |
| 10 | 25.391 | 8.373 | 5.957 | 1.723 |





## 4 Discussion

### 4.1 Key parameters and their physical effects

Several parameters have been identified in Sections 3.3 and 3.4 as having a significant impact on the simulated SIC and SIV budgets in the Southern Ocean. In this section we present how these parameters specifically act on the SIC and SIV budget

by looking at the impact of parameter changes on the cumulative distribution function (CDF) in the PAWN method.

Considering the performance of the GP emulator (Fig. B6) as well as the number of sensitive parameters (Fig. 6), the area integral of $res$ component in the SIC budget in spring and the area integral of $adv$ component in the SIV budget in winter have been selected here as examples to be discussed. Figs. 9 and 10 show how the CDF of the model output changes as one parameter is fixed to vary across a range of values, and other parameters varied freely.


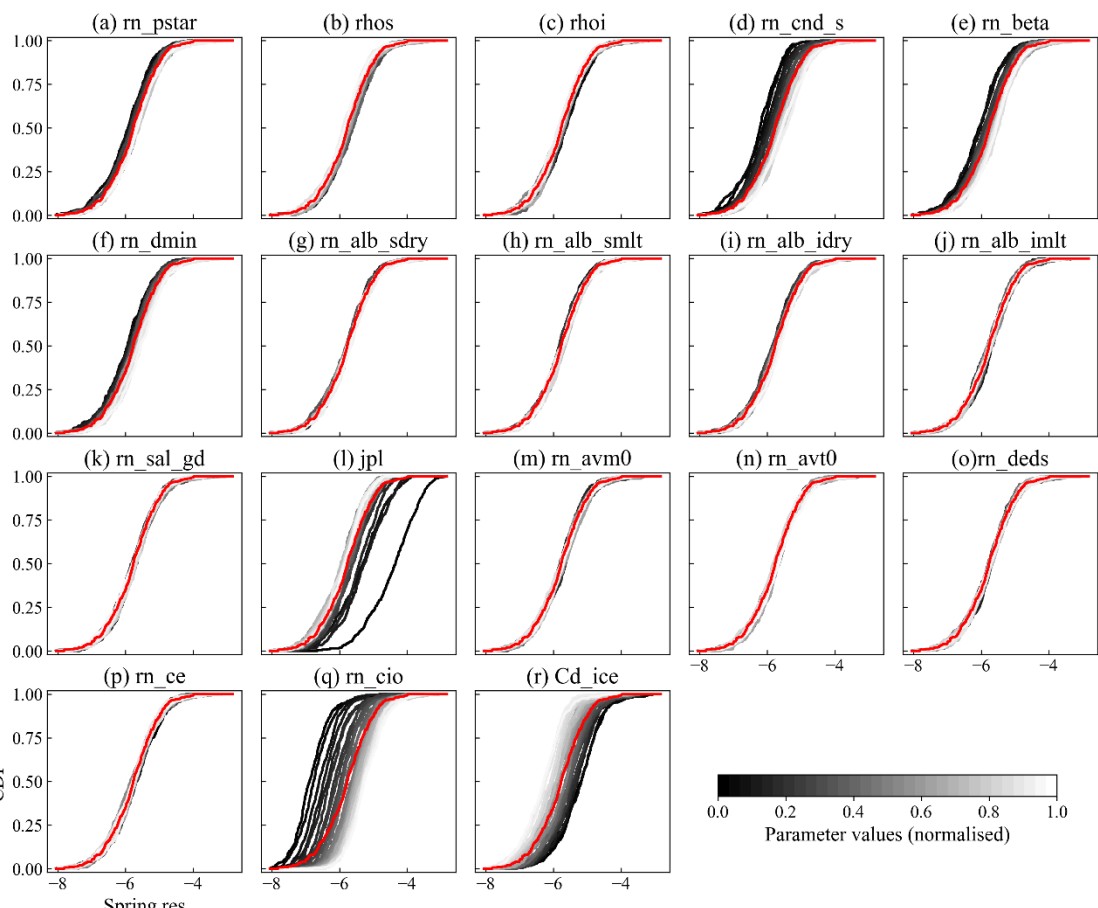

**Figure 9.** Cumulative distribution function (CDF) of the area integral of the $res$ component in the spring SIC budget (cf. Fig. 3). Red lines are the unconditional CDF for the ensemble of 449 model runs, and the grey lines stand for conditional CDF at different fixed values of parameters calculated by the GP emulator. The units of the x-axis are $10^6 \ km^2$.





Since the low thermal conductivity of the snow reduces the heat transfer from the bottom of the ice to the atmosphere, it reduces the ice growth rate (Fichefet et al., 2000; Lecomte et al., 2013), and therefore leads to less freezing inside the ice pack, and *res* moves more towards negative values (Fig. 9d). The reduction in freezing due to the reduction in snow thermal conductivity is more pronounced in winter (Fig. 6) and the SIV budget simulation is more sensitive to this parameter than the SIC budget, as it is primarily affecting the vertical ice growth.

The rn_beta and rn_dmin are the two parameters that determine the minimum floe diameter of sea ice, and their decrease implies a decrease in sea ice floe sizes, which promotes the lateral melting (Lüpkes et al., 2012). Consequently, in contrast to the reduction of rn_cn_s which inhibits ice freezing, rn_beta and rn_dmin lead to more negative values of *res* (Fig. 9e-f) by promoting sea ice melting at low-latitude regions (Fig. 3). Furthermore, this effect is greater in summer than in spring and plays a weak role in winter (Fig. 6), which fits well with the magnitude of the SIC reduction in the *res* column in Fig. 3,

although it is not the only process affecting SIC.

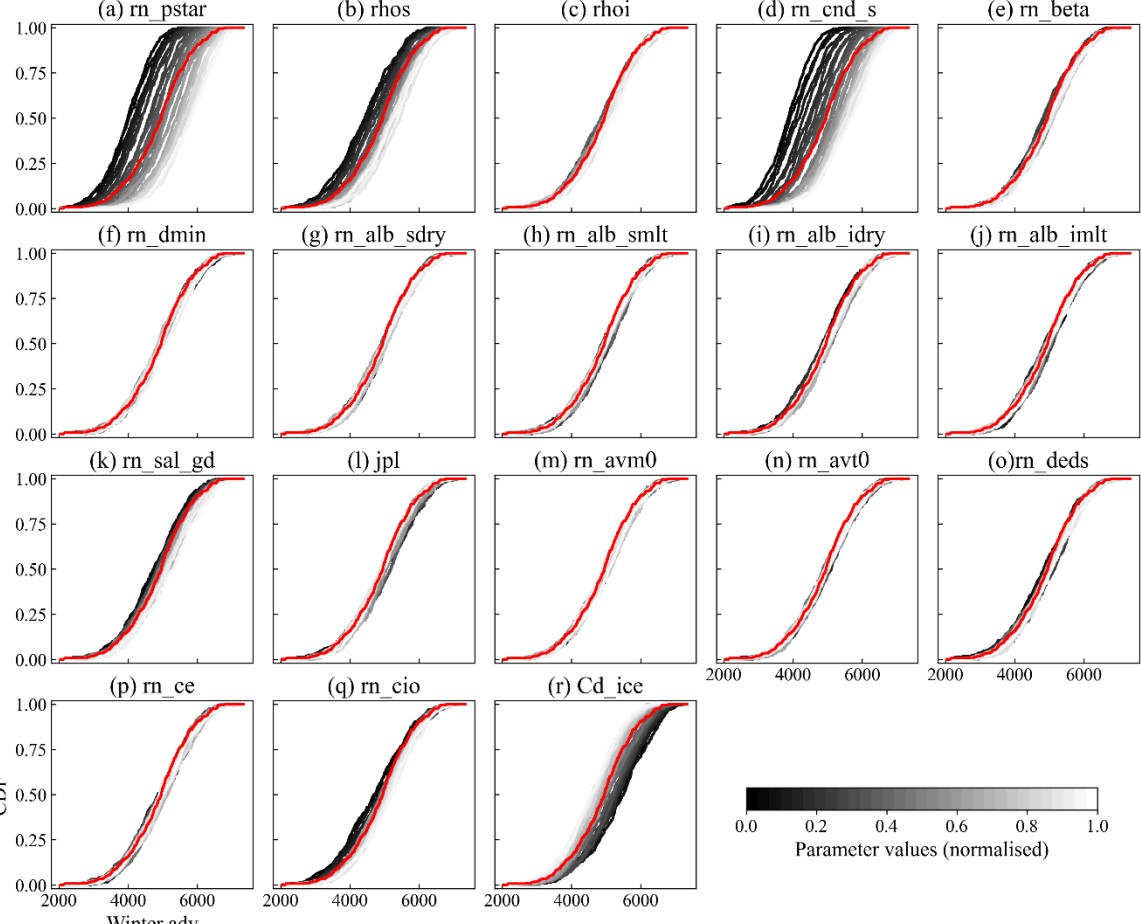

**Figure 10.** As Fig. 9, but for the area integral of $adv$ component of winter SIV budget. The units of the x-axis are $10^3\ km^3$.



Compared to rather continuous looking variations in CDFs of other parameters, the variation in CDFs due to changes in jpl is more dispersed (Fig. 9l), with several lines being clearly outliers, which were checked to match jpl=1. This is because the multi-category sea ice thickness takes into account the subgrid-scale variations in sea ice properties (Thorndike et al., 1975; Massonnet et al., 2019; Moreno-Chamarro et al., 2020) and is therefore significantly different from the single thickness category (jpl=1). For instance, the presence of thin sea-ice categories in multi-category sea-ice schemes allows for greater melt rates compared to a single-category scheme (Uotila et al., 2017).

The ice-ocean drag coefficient and the air-ice drag coefficient should be discussed jointly, as the sea-ice drift velocity is related to the Nansen number $Na = \sqrt{\rho_a C_a / \rho_w C_w}$, where $\rho_{a/w}$ and $C_{a/w}$ are air/water density and air-ice/ice-ocean drag coefficient. The Fig. 9q and 9r illustrate that a decrease in $C_a/C_w$ leads to a larger $res$, which has two possibilities, either sea ice melt is inhibited or freezing is intensified, by assuming that sea ice deformation is comparably small (Holland and Kwok, 2012). Since the solution of free sea ice drift (Leppäranta, 2011, Chapter 6.1.1) indicates that the decrease in $C_a/C_w$ leads to a decrease in sea ice velocity, we argue that this causes a more limited transport of sea ice to low-latitude region, leading to the inhibited melting (see spring $adv$ and $res$ in Fig. 3).

With the exception of rn_cnd_s, rn_cio and Cd_ice, whose physical effects have been elucidated, the $adv$ term in the winter SIV budget is also sensitive to rn_pstar (Fig. 10a). This can be explained by the fact that the weaker ice is more easily to deform and increase ice thickness, leading to a smaller drift speed and therefore results in a smaller absolute value of the area integral of $adv$ or $div$. This is also true in spring (Fig. 6), as ice drift speeds are greater in winter and spring compared to other seasons during the period of this study (not shown but similar to, e.g., Holland et al., 2016) and making the ridging of weak ice more pronounced.

For the NEMO4.0-SI[3], the snow thickness on sea ice is determined by the snow density as the solid precipitation equivalent which is determined by atmospheric reanalyses, and other factors affecting the snow depth (e.g., wind packing, windblown snow lost to leads, etc.; Petty et al., 2018) that are not included (NEMO Sea Ice Working Group, 2019). When the snow density decreases in the model, the snow thickness increases, thereby reducing the heat exchange between the ice and the atmosphere, which in turn limits the vertical increase in sea ice thickness. Thus, for the SI[3] model, the effect of reducing snow thickness and reducing snow thermal conductivity on the simulation of sea ice thickness is equivalent. This is the reason why the $res$ term in the SIV budget always shows a similar high sensitivity to rn_cnd_s and rhos (Fig. 6). These two parameters have the greatest influence on the total SIV and thus also on the area integral of the $adv$ during autumn and winter, the seasons when sea ice vertical growth is most pronounced. When sea ice thickening is limited, the value of SIV itself becomes smaller, resulting in a smaller area integral for $adv$ (Fig. 10b).

However, of the seven parameters discussed above that have an impact on the SIC budget, only two drag coefficients play a critical role to the RMSE of simulated and observed SIC budget, followed by the weak effect of sea ice strength (Fig. 7). This means that while adjusting rn_cnd_s has an impact on the simulation of SIE (Urrego-Blanco et al., 2016) and may improve the SIE seasonal cycle to be closer to observation (Lecomte et al., 2013), it does not make the model's simulation of





the SIC budget any more realistic. In addition, although the remaining parameters display sensitivity during the summer months (bottom row in the Fig. 7), the robustness of this result is not guaranteed given the already low level of RMSE in the summer and the mediocre performance of the GP emulator (bottom row in the Fig. B7).

### 4.2 Interactions between the parameters

Using the second order sensitivity indices provided by the Sobol method, the interaction between the parameters can be further explored. We have added some vertical connector lines in Figs. 6 and 7 to indicate that a simultaneous change in two parameters question has a significant impact. Not surprisingly, the interconnection of the ice-ocean and the air-ice drag coefficients causes their simultaneous changes to have the greatest impact on the advection metric in both SIC and SIV budgets, especially in winter and spring, the two seasons with the largest sea ice speeds. Furthermore, for the SIV budget, the contribution of its advection term to SIV change is also sensitive to the simultaneous changes in rn_cnd_s and rn_cio in autumn. This makes sense, considering that sea ice starts to grow vertically in autumn and that the advection is significantly affected by the ice-ocean drag coefficient (Fig. 6). However, rn_cnd_s does not interact with any drag coefficient in winter, when ice vertical grow is also rapid (Fig. 2c), thus the interaction in autumn remains somewhat uncertain due to the GP emulator does not perform very well for $adv$ in the autumn SIV budget (r=0.961).

The ratio between the ice-ocean and the air-ice drag coefficients continues to dominate the sensitivity of the four RMSE metrics as the sea ice velocity is controlled by $C_a/C_w$ (Fig. 7). Although the GSA results also show some sensitivity to ice strength, there is little interaction between this parameter and the two drag coefficients in the SIV budget, except for the $adv$ term in summer. Despite this, considering that the $adv$ RMSE$_{SICB}$ itself fluctuates very little in summer and the GP emulator is not a perfect performer, there is uncertainty in this result. Fig. 7 also shows that the $dadt$ RMSE$_{SICB}$ is sensitive to simultaneous changes in rn_beta and rn_cio in the autumn, which we argue may be an error introduced by the poorer performing GP emulator (r=0.915) as the lateral ice growth process is independent with the ice floe size in NEMO4.0-SI[3] (NEMO Sea Ice Working Group, 2019).

### 4.3 Recommended set of parameters

Fig. 2 highlights the SIE, SIA and SIV seasonal cycles of the three experiments that performed best in the mean RMSE$_{SICB}$ (as listed in Table 3). An interesting thing is that although these three experiments used rn_cio/Cd_ice values that were clearly above/below the standard values, they all exhibit SIE and SIA seasonal cycles that are very close to the model ensemble mean and the CTRL. The EXP397, which is the best performing one, has a SIV seasonal cycle that almost overlaps with the ensemble mean, while the second and third best are both close to the CTRL. This evidence again suggests that even if the realistic SIE is modelled, there is no guarantee of a reasonable SIC budget (Uotila et al., 2014; Nie et al., 2022), e.g., the mean RMSE between the CTRL and observed SIC budgets is similar compared to the other experiments, whereas the SIE seasonal cycle of the CTRL is very realistic.





On the other hand, even the optimal set of parameters recommended in this study (EXP397) would only reduce the $dadt$, $adv$, $div$ and $res$ RMSE$_{SICB}$ by about 2%, 5%, 8% and 10% respectively, which is a rather modest impact. This indicates
that the accurate modelling of the SIC budget does not appear to be possible by simply changing the atmospheric forcing product or tuning ocean model's parameters, as the atmospheric forcing itself is systematically biased (Barthélemy et al., 2018). As shown in Fig. B8, all model ensembles have similarly shaped ice-speed seasonal cycles that all differ significantly from observations, meaning that adjusting the parameter values alone will not correct errors caused by biases in the atmospheric forcing. Nevertheless, the parameter sets in Table 3 can be confidently recommended to NEMO4.0-SI[3]
modelers to optimize the Southern hemispheric sea ice in the ORCA2 grid, provided that DFS5.2 is used as the atmospheric forcing.

## 5 Conclusions

To investigate the impacts of model parameter uncertainty on sea ice budgets in the Southern Ocean, we drove the NEMO4.0-SI[3] ice-ocean coupled model with DFS5.2 atmospheric forcing and simultaneously adjusted 18 potentially critical
model parameters and generated the model ensemble with a size of 449. Preliminary diagnostics of the model output for the SIE and SIA seasonal cycles revealed that the model results are generally reasonable, as the ensemble model mean being very close to observations. The ensemble model mean SIC budget shows the basic characteristics of the observed SIC budget, although differing a lot in details, and the adjustment of the parameters indeed leads to a certain degree of perturbation of the SIC and SIV budgets, which sets the stage for the sensitivity experiments that followed.
Benefiting from the overall excellent performance of the GP emulator, GSA was carried out with adequate computational resources. The results show that the contribution of the modelled advection to the changes in SIC is very sensitive to ice strength, ice-ocean and air-ice drag coefficients from autumn to spring, and to snow thermal conductivity in summer, followed by two other parameters related to lateral melting as well as the ice-ocean drag coefficient. Additionally, the $res$ term in summer is very sensitive to the number of ice categories, which is attributed to the significant difference in sea ice
melt rates between single and multi-category sea ice categories. In addition to several parameters that have an impact on the simulation of the SIC budget, the SIV budget also shows a high sensitivity to snow density. However, considering the simple approach to snow in the current NEMO4.0-SI[3] model (e.g., one layer and the effect of windblown is not taken into account, etc.), the effects of snow density and snow thermal conductivity on sea ice thickness are largely equivalent.

The sensitivity of the RMSE$_{SICB}$ to 18 parameters was assessed. Overall, the ice-ocean and air-ice drag coefficients are the
most important ones, followed by ice strength. Moreover, there are other parameters that significantly affect RMSE$_{SICB}$ during the summer months, but since RMSE$_{SICB}$ values are inherently small during the summer months, we consider the effects of these parameters on the RMSE$_{SICB}$ to be negligible. Based on these results, we recommend 10 combinations of ice-ocean drag coefficient, air-ice drag coefficient and ice strength that can be safely used for the DFS5.2 driven NEMO4.0-SI[3] model with the ORCA2 grid. The recommended combinations of these parameters allow the simulations of near-observed



SIE and SIA seasonal cycles, as well as similar SIV seasonal cycles with the CTRL experiment and, more importantly, resulting in a more realistic SIC budget compared to the standard parameters.

Apart from the success of the GP emulator, another reason why the GSA results are considered reliable is that the two GSA methods used in this paper show a high degree of consistency in the identification of key parameters. Nevertheless, we recommend that it is necessary to use two or more GSA methods together to target same problem, as variance-based Sobol method and density-based PAWN method each have their own characteristics and can be cross-referenced and complement each other, which has also been revealed in other studies (e.g., Pianosi and Wagener, 2015; Zadeh et al., 2017; Mora et al., 2019).

There are at least two limitations in this study, the first is that we selected the area integral of $adv$ and $res$ as metrics, and although they can be used as proxies for the total contribution of dynamical and other processes to sea ice change respectively, the local biases may counteract and affect the integrals. We therefore complemented this with another set of metrics using the $RMSE_{SICB}$. The second limitation stems from the fact that uncertainties in observations cannot be accurately assessed and the observed budgets were simply referred to as the "true", which could be re-evaluated after more accurate observations become available, or when the uncertainties in observed ice motion can be more accurately estimated.

In summary, the key to reproducing a realistic SIC budget for an ice-ocean coupled model driven by atmospheric reanalysis is to simulate realistic sea ice velocities, which undoubtedly remains a challenge. It would be very useful to correct the biases in the atmospheric reanalysis, and the model could then be further optimised by adjusting several key parameters identified in this study.

*Code and data availability.* The model code for NEMO4.0-SI[3] is available from the NEMO website (https://www.nemo-ocean.eu/, last access: 1 March 2022). The parameter sets, configuration files and scripts for running NEMO4.0-SI[3] are archived on https://doi.org/10.5281/zenodo.6780342 (Nie, 2022). The atmospheric forcing was provided by the DRAKKAR consortium through the following link: https://ige-meom-opendap.univ-grenoble-alpes.fr/thredds/catalog/meomopendap/extract/FORCING_ATMOSPHERIQUE/DFS5.2/ALL/catalog.html (last access: 22 February 2022). The NOAA/NSIDC Climate Data Record of Passive Microwave Sea Ice Concentration, Version 4 (Meier et al., 2021) data can be downloaded from National Snow & Data Center (https://nsidc.org/) by registering for an EarthData account. The KIMURA ice drift data are available from the authors on request. The GPy code is available here: https://github.com/SheffieldML/GPy (last access: 1 March 2022). The SAFE Toolbox used for implement the PAWN method is available here: http://bristol.ac.uk/cabot/resources/safe-toolbox/ (last access: 11 April 2022).

*Author contributions.* PU, YFN and XQL designed the study. YFN and PU run the NEMO4.0-SI[3] model. CKL and YFN built the GP emulator. Data analysis was performed by YFN, PU, BC and FBD. The first draft of the manuscript was written




by YFN, PU, MV and all authors commented on previous versions of the manuscript. All authors read and approved the final manuscript.

*Competing interests.* The authors declare that they have no known competing financial interests or personal relationships that could have appeared to influence the work reported in this paper.

*Acknowledgements.* The authors acknowledge CSC – IT Center for Science, Finland, for HPC computational resources.

*Financial support.* PU was supported by the Academy of Finland (Project 322432), and the European Union's Horizon 2020 research and innovation framework programme under Grant agreement no. 101003590 (PolarRES project). XQL was supported by the National Natural Science Foundation of China (Grant No. 42076011 and Grant No. U1806214) and YFN was supported by Scholarship from China Scholarship Council (CSC. 202006330054).

**Appendix A: Global sensitivity analysis**

Two different kinds of GSA methods were performed here, as only one may not adequately bring out all the characteristics (Baki et al., 2022; Pianosi et al., 2015). The first one is the variance-based sensitivity analysis, which is also referred to as Sobol indices (Sobol, 2001). Suppose the relationship between model output $Y$ and parameter sets $X$ is $Y = f(X)$, where $X_i \in [0,1]$, $i = 1,2,\ldots,p$, and it can be decomposed as (Sobol, 1990):

$$Y = f_0 + \sum_{i=1}^p f_i(X_i) + \sum_{i<j}^p f_{ij}(X_i, X_j) + \cdots + f_{1,2,\ldots,p}(X_1, X_2, \ldots, X_p), \tag{A1}$$

where $f_0$ is a constant, $f_i$ and $f_{ij}$ are functions of $X_i$ and $X_{ij}$ respectively, and so on. Then the $i^{th}$ parameter's first-order indices ($S_i$) and total-effect index ($S_{Ti}$) are estimated as (Sobol, 2001; Saltelli et al., 2010):

$$S_i \approx \frac{\frac{1}{N}\sum_{j=1}^N f(\boldsymbol{B})_j \left(f(\boldsymbol{X}_B^i)_j - f(\boldsymbol{X})_j\right)}{\sum_{i=1}^p V_i + \sum_{i<j}^p V_{ij} + \cdots + V_{12\ldots p}}, \tag{A2}$$

$$S_{Ti} \approx \frac{\frac{1}{2N}\sum_{j=1}^N \left(f(\boldsymbol{X}_B^i)_j - f(\boldsymbol{X})_j\right)^2}{\sum_{i=1}^p V_i + \sum_{i<j}^p V_{ij} + \cdots + V_{12\ldots p}}, \tag{A3}$$

where $V_i = Var_{X_i}\left(E_{\boldsymbol{X}_{\sim i}}(Y|X_i)\right)$, $V_{ij} = Var_{X_{ij}}\left(E_{\boldsymbol{X}_{\sim ij}}(Y|X_i, X_j)\right) - V_i - V_j$, and so on, the $\boldsymbol{X}_{\sim i}$ indicates the set of all 570 parameters except $X_i$. The matrix $\boldsymbol{B}$ is a $N \times p$ matrix generated by sampling the parameter space with the LHS method and used as a "perturbation matrix". $N$ denotes the number of model simulations. The matrices $\boldsymbol{X}_B^i$, $i = 1,2,\ldots,p$ are obtained by replacing the $i^{th}$ column of $\boldsymbol{X}$ with the same column of $\boldsymbol{B}$.

The other GSA method named PAWN (Pianosi and Wagener, 2015) is a density-based method, in which sensitivity is assessed by quantifying the effect of parameter changes on the cumulative distribution function (CDF) of the model output $Y$.





In brief, the distance between the CDF of $Y$ obtained from the control simulation (i.e., unconditional CDF) and the CDF of
the output perturbed by changing the parameters (i.e., conditional CDF) is calculated by the Kolmogorov-Smirnov statistic
(KS):

$$KS(X_i) = \max_{Y} \left| F_Y(Y) - F_{Y|X_i}(Y) \right|, \tag{A4}$$

where $F_Y(Y)$ is the unconditional CDF and $F_{Y|X_i}(Y)$ is the conditional CDF with the fixed $X_i$. Since the KS statistic may vary

due to $X_i$ taking different values, the PAWN index $T_i$, which indicates the sensitivity of $Y$ to $X_i$, is then obtained by
considering a statistic (e.g., maximum or median) over all possible $X_i$:

$$T_i = \operatorname*{stat}_{X_i}[KS(X_i)]. \tag{A5}$$





## Appendix B: Supplementary Figures

**Fig. B1.** Seasonal mean of sea ice concentration (SIC) budget components for 2008-2017, calculated based on satellite-derived sea ice velocity (Kimura et al., 2013) and SIC (Meier et al., 2021) observations. The positive value stands for the SIC increase and the negative value for the decrease.





**Fig. B2.** (a-b) Observed (NOAA/NSIDC Climate Data Record of Passive Microwave Sea Ice Concentration, Version 4; CDR) and model ensemble mean February SIC climatologies (only SIC > 15% are shown), (c) standard deviation of all model runs. (d-f) The same as (a-c) but for September.



**Fig. B3.** (a) Ensemble model mean February sea ice thickness climatologies (only SIC > 15% are shown) and (b) the standard deviation. (c-d) The same as (a-b) but for September.



**Fig. B4**. Standard deviation of seasonal SIC budget components for the ensemble of 449 model runs.



**Fig. B5**. As Fig. B4, but for sea ice volume (SIV) budget.





**Fig. B6**. Validation results of the best Gaussian process (GP) emulators for each of the four metrics (area integrals of $adv$ and $res$ components in SIC and SIV budgets) selected by the 10-fold cross-validation. Each subplot consists of 449 error bars and a 1:1 line, and Pearson correlation coefficients are also listed. Each metric has been normalized (scaled to [0, 1] using the difference between the maximum and minimum values of the simulation) for better presentation.



**Fig. B7**. As Fig. B6, but for the root-mean-square error between SIC budget components of the simulation and the observation ($RMSE_{SICB}$).





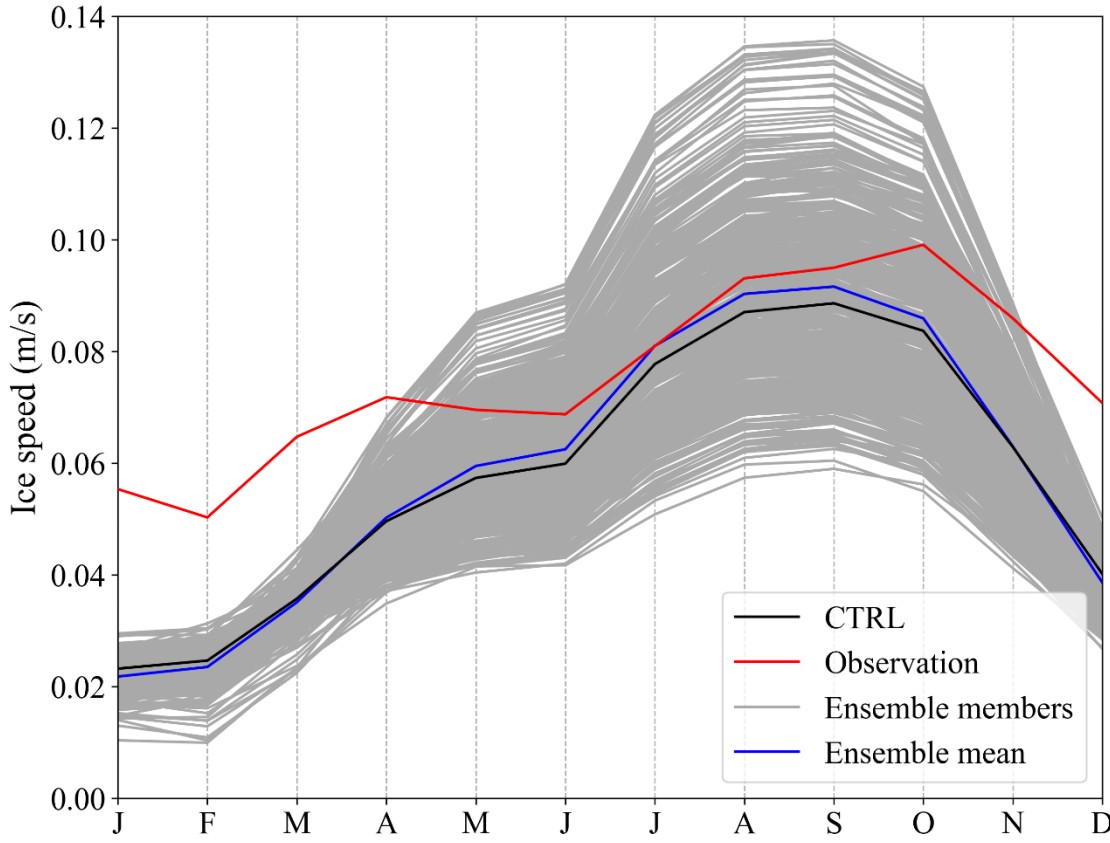

**Fig. B8**. Sea ice speed seasonal cycles for the observation (Kimura et al., 2013) and simulations, over 2008-2017. The simulated sea ice velocities are first interpolated onto the KIMURA data grid, then the spatial average of the ice speed is calculated in the areas where observations are available.





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
