# Peer review of "Sensitivity of NEMO4.0-SI3 model parameters on sea ice budgets in the Southern Ocean"

_Geoscientific Model Development, 2022_

## Referee Comment (RC2)

[referee-annotated manuscript omitted]

---

## Author Comment (AC1)

Reviewer #1: This submission addresses relevant and timely scientific sea ice modelling questions – in particular, how different model parameters, and combinations of model parameters, influence both sea ice budgets in the Southern Ocean as well as comparison with satellite data. The sea ice components of global climate models tend to rely on parameterizations dominated by Arctic studies. This paper is a new look at parameterizations targeting Antarctic sea ice, and recommends ten new combinations of parameters for the NEMO4.0-SI3 sea ice model that would result in better comparisons with satellite-based observations of Southern Hemisphere sea ice area and extent. The paper is well written overall, with methods and assumptions clearly outlined.

We thank you for the constructive comments on the earlier version of the manuscript. We have revised our manuscript following the comments, in the following we answer each specific point (in blue).

The submission will be stronger with some revisions including:

1. Set the broader context. This submission explores sea ice parameterizations primarily aimed at reducing RMSE between the model output and satellite observations for total Southern Hemisphere Sea Ice Extent and Sea Ice Area. These are very limited metrics for model performance – e.g. Notz, 2014 and Notz, 2015. The experiments in this paper are thoughtful and provide interesting insight into sea ice models yet the results presented here are not presented within the larger context, and only look at a very limited metric (climatological mean SH SIA, SIE). Other metrics that are valuable include variability, trends, and particularly for the Antarctica regional means, variability and trends. How would the recommended parameterizations, for example, impact (or not) NEMO's future scenario simulations? Do they impact variability and trends? Are the improvements to SH SIA, SIE also seen in all regions or are they regionally different? (or if beyond the scope of this paper some mention and discussion…)

We totally agree with the reviewer that SIE and SIA are very limited metrics for assessing the model, and other valuable metrics should include (regional) variability

and trends, as revealed by Notz (2014) and Notz (2015). This was precisely the starting point of this paper, i.e., to use more physically meaningful metrics than SIE and SIA for model evaluation and optimisation. Our chosen metric is the sea ice budget, which decomposes sea ice variability into advection, divergence and other processes (mainly thermodynamic). Therefore, our aim in this study is (Lines 73-75) "to quantify the sensitivity of the Southern Ocean SIC and sea ice volume (SIV) budgets to key parameters in a coupled ocean-sea ice model by constructing a GP emulator, and furthermore, to verify whether the model parameters can be adjusted to obtain near-realistic SIC budget components." Moreover, we believe that the first three sentences of the abstract clearly communicate this point.

Indeed, in Fig. 2 we show the climatological mean SH SIA, SIE, which, as mentioned, are the primary metrics. On the one hand, these metrics verified that the ocean-sea icemodel forced by the atmospheric reanalyses we used was working reasonably well, and more importantly, provided good evidence that our approach to optimise parameters by reducing the RMSE between the modelled and observed SIC budget makes good sense, i.e. although the default NEMO4-SI3 parameters can already produce reasonable SIE and SIA seasonal cycles, their physical processes are likely to differ significantly from the reality.

As for the model results, our analysis is unfortunately limited to the period 2008-2017, which is only a decade, and not sufficient to support investigations of the impact of these parameters on inter-decadal sea ice trends. In terms of regional differences, we have added Fig. B6 which illustrates that the optimisation improves the SIC budgets in all regions (Line 511).

[Figure]

**Fig. B6.** As Fig. 7, but the RMSE of each SIC budget term is averaged over four seasons and counted separately in each Southern Ocean sector. The vertical dotted line marks the demarcation of each sector. AB=Amundsen-Bellingshausen Seas.

2. Some discussion of why these parameterizations are better in the SH and some differences between Antarctic sea ice and Arctic sea ice and why these differences might lead to these different parameterizations (I am assuming current NEMO parameterizations are based on Arctic work).

We have added Fig. B8 to illustrate that several of the parameter sets that are associated with top performing Southern Ocean SIC budgets also perform better in the Arctic SIE simulations compared to the NEMO4-SI3 default values (Lines 521-523), which at least gives us more confidence that these parameter sets are reliable. However, it must be re-emphasised that these parameters are highly dependent on the atmospheric forcing data used.

The added text reads (Lines 521-527) "In addition, Fig. B8 shows that the recommended parameter sets also provide some improvements in the Arctic SIE and SIA simulations compared to the default parameters, as reflected by more sea ice in summer months, which is closer to observations than in the CTRL experiment.

However, given that SIE and SIA are limited metrics (Notz, 2014; Notz, 2015) and that the key parameters affecting sea ice simulations may not be the same between the northern and southern hemispheres due to the vast geographical differences (e.g. ocean and land locations, atmospheric and oceanic circulations), whether these parameter sets, which perform well in the Southern Ocean SIC budget, can be safely applied to the Arctic merits further investigation."

[Figure]

**Fig. B8.** Simulated monthly climatologies of Arctic (a) sea ice extent (SIE) and (b) area (SIA) from 2008 to 2017, ensemble model means and results from four sets of experiments of interest are also highlighted. The SIE and SIA calculated from the CDR, AMSR-E/AMSR2, CERSAT and OSISAF are used as references in the form of mean ± one deviation.

3. These parameterizations are determined from current conditions. Any thoughts as to whether or not they would be expected to be constant and/or changing in a warming world?

Main principle is that parameterizations describe unresolved physics of models, therefore also these parameter values are likely to change under changing circumstances. The direction and amount of the change varies from parameter to parameter and non-linearities in the climate systems make the estimation of changes particularly tricky. For example, the air-ice drag coefficient would change due to changing sea ice and snow

surface roughness and changing atmospheric stability. In a warming world the first order effect is intensified hydrological cycle and precipitation which could be more in liquid form potentially reducing surface roughness and air-ice drag. At the same time, the atmospheric boundary layer could become less stable potentially increasing the air-ice drag coefficient. The estimation of the net effect is not easy. And there are regional and seasonal differences in terms of parameter responses. In summary, one could expect the parameter values to change.

We added the following sentence in the revised manuscript (Line 568) "The recommended parameter sets are determined from current conditions and one could expect their optimal values to change in a warming world."

4. In the first paragraph in the Introduction, the authors discuss how climate models in general do not capture the observed trends in SH sea ice. While this is true, at no point in this paper are the parameterizations discussed in light of the trends! The parameterizations are compared only to the climatological mean SIA, SIE – not the variability or the trends (or regionality). The spread in representation of the mean annual cycle of SIE is quite large between CMIP6 models (e.g. Roach et al., 2020), however there are climate models that capture the climatological annual cycle of SH SIE. Clarify the introduction a bit to match the research and results presented.

We agree that the interpretation of sea ice trends in the first paragraph of the introduction may distract the reader from the fact that in this paper we are not optimising model parameters based on sea ice trends. In fact, our focus is on the sensitivity of the Southern Ocean sea ice budgets (which we believe are more valuable metrics than SIA and SIE) to the parameters, and thus we propose some reliable parameter sets. To make this more clear, we modified the first paragraph to (Lines 24-31):

"Several state-of-the-art climate models have successfully simulated the near-realistic annual cycle of sea ice area (SIA) (Holmes et al., 2019), but they typically still fail to capture the observed sea ice variability and trends (Zunz et al., 2013; Turner et al., 2013; Shu et al., 2015; Shu et al., 2020). This implies that standard metrics

commonly used for model evaluation, such as sea ice extent (SIE), SIA and total volume (SIV), are rather rudimentary and of limited use in improving the model skill (Notz, 2014; Notz, 2015), and better metrics are needed to optimise models."

It is true that some CMIP5 and CMIP6 models could capture the realistic SIE seasonal cycles, but not the sea ice trends. Furthermore, it has also been shown in several studies that these realistic SIEs are also caused by large biases in the dynamic and thermodynamic contributions to sea ice variability that cancel out each other (e.g., Uotila et al., 2014; Lecomte et al., 2016; Holmes et al., 2019). The relationship between sea ice budgets and sea ice trends is a very interesting topic and our ongoing work, but beyond the scope of this paper.

5. It may be interesting to add three panels to figure 2 showing variability (STD) of each of these as well……….or not if beyond the scope..

Thanks for your valuable suggestion, but it is beyond the scope of this study to investigate how the accuracy of sea ice budget simulations affects the modelling of sea ice variability and trends. However, this is an interesting topic we are now working on.

6. Passive microwave images will lead to underestimates of SIC in thin ice regions. Any thoughts to whether or not this influences how one compares model output to satellite (you only consider areas of SIC 15% and higher. What about regions where model output is > 15% SIC and sea ice thicknesses less than 5 cm or 5-20cm where satellite observations underestimate SIC?

We only calculate the sea ice budget for grids with SIC > 15% because sea ice velocity observations are more reliable in this interval, and therefore allows for a fair comparison of modelled and observed SIC budgets (Holmes et al., 2019). At the same time the effect of observed underestimates in the region of SIC < 15% on the SIC budget calculation is excluded, and in any case the SICB is rather insensitive to such SIC underestimates (Holland and Kimura, 2016).

In addition, when comparing simulated and observed SIAs (Fig. 2), the uncertainty of satellite observations at SIC <15% is worth considering. To address this, we used

multiple observational products and plotted the range of observational uncertainty in Fig. 2, where model's significant overestimation of the August to October SIA is shown being somewhat weaker when the observational uncertainty is taken into account compared to if only the CDR product is used.

[Figure]

**Figure 2.** Simulated monthly climatologies of (a) sea ice extent (SIE), (b) area (SIA) and (c) volume (SIV) from 2008 to 2017, ensemble model means and results from four sets of experiments of interest are also highlighted. The SIE and SIA calculated from the CDR, AMSR-E/AMSR2, CERSAT and OSISAF are used as references in the form of mean ± one deviation.

The submission is well written in general however there are some times when it is a bit unclear due to grammar or word choice. I found some minor changes along these lines and I believe the manuscript would benefit from the help of a skilled editor for language

word choices, etc. Here are some suggested minor changes:

Line 18: change "sensitivity" to "sensitive"

Revised.

Line 27: change "association" to "teleconnections"

The sentence in the original manuscript where this word was located has been removed.

Line 93: This is confusing. I believe you mean "number of sea ice thickness categories is 5" and I have no idea what "2 and 1 layers of ice and snow" means. How can a 5-thickness categories for sea ice only have 2 layers? Or one? Guessing just 1 layer of snow on top of sea ice?

We have rephrased this sentence to make it clearer and it now reads (Line 92) "The default number of sea ice thickness categories is 5, with each category having two vertical layers of ice and one layer of snow on top of ice."

Line 110: change "marginal regions" to "marginal sea ice regions" (and define "marginal"…15-85% SIC? Or?)

We note that the expression "marginal regions" is misleading and naturally reminds the reader of the "marginal sea ice regions", when in fact we are trying to express the "edge of the sampling interval". We have now revised this phrase in Line 111.

Table 1: jpl = "number of ice thickness categories" I believe? Or? And are these set – in other words, do you change not only the number of ice thickness categories but also the category boundaries? Or just the number?

Only the number of ice thickness categories were tuned in this study, and the position of boundaries were prescribed by default by using a fitting function that following Lipscomb (2001) in NEMO4-SI3 default. We have now changed the description of jpl in Table 1 to "Number of ice thickness categories".

**Table 1.** The 18 parameters investigated, including their realistic ranges taken from the listed references.

| Category | Symbol | Description and unit | Low | Standard | High | Reference |
|---|---|---|---|---|---|---|
| Ice/snow | rn_pstar | Ice strength parameter [N/m2] | 5.00E+03 | 2.00E+04 | 3.50E+04 | Massonnet et al. (2014) |
| | rhos | Snow density [kg/m3] | 130 | 330 | 530 | Massom et al. (2001) and Warren et al. (1999) |
| | rhoi | Ice density [kg/m3] | 880 | 917 | 940 | Timco and Frederking (1996) |
| | rn_cnd_s | Thermal conductivity of the snow [W/m/K] | 0.1 | 0.31 | 0.5 | Maykut and Untersteiner (1971) and Lecomte et al. (2013) |
| | rn_beta | Coefficient beta for lateral melting parameter | 0.2 | 1 | 1.8 | Lupkes et al. (2012) |
| | rn_dmin | Minimum floe diameter for lateral melting parameter [m] | 2 | 8 | 14 | Lupkes et al. (2012) |
| | rn_alb_sdry | Dry snow albdo | 0.85 | 0.85 | 0.87 | Perovich et al. (2002) and Brandt et al. (2005) |
| | rn_alb_smlt | Melting snow albdo | 0.72 | 0.75 | 0.82 | Perovich et al. (2002) and Brandt et al. (2005) |
| | rn_alb_idry | Dry ice albdo | 0.54 | 0.6 | 0.65 | Perovich et al. (2002) and Brandt et al. (2005) |
| | rn_alb_imlt | Melting ice albdo | 0.49 | 0.5 | 0.58 | Perovich et al. (2002) and Brandt et al. (2005) |
| | rn_sal_gd | Restoring ice salinity, gravity drainage [g/kg] | 4 | 5 | 7.5 | Nakawo and Sinha (1981) |
| | jpl | Number of ice thickness categories | 1 | 5 | 30 | Massonnet et al. (2019) |
| Ocean | rn_avm0 | Eddy viscosity [m2/s] | 1.00E-05 | 1.20E-04 | 1.50E-04 | Williamson et al. (2017) |
| | rn_avt0 | Eddy diffusivity [m2/s] | 1.00E-06 | 1.20E-05 | 1.50E-05 | Williamson et al. (2017) |
| | rn_deds | Magnitude of the damping on salinity [mm/day] | -20 | -166.67 | -180 | NEMO System Team (2022) |
| | rn_ce | Magnitude of the mixed layer eddy | 0.04 | 0.06 | 0.1 | NEMO System Team (2022) |
| Coupling | rn_cio | Ice-ocean drag coefficient | 2.00E-03 | 5.00E-03 | 8.00E-03 | Massonnet et al. (2014) |
| | Cd_ice | Air-ice drag coefficient | 8.00E-04 | 1.40E-03 | 2.00E-03 | Massonnet et al. (2014) |

Reference

Lipscomb, W. H. (2001). Remapping the thickness distribution in sea ice models. Journal of Geophysical Research: Oceans, 106(C7), 13989-14000.

Line 332 change "ice category number" to "number of ice thickness categories"

Fixed.

Line 424: add "SIC" before "CDFs"

Since the full name of the "CDFs" here is "CDFs of the area integral of the res component in the spring SIC budget " but not the "SIC CDFs", we abbreviate this here as "CDFs", and the reader can easily see what this CDFs stands for from the figure caption of Fig. 11.

References

Notz D. 2014 Sea-ice extent and its trend provide limited metrics of model performance. Cryosphere 8, 229–243. (doi:10.5194/tc-8-229-2014)

Notz, D.: How well must climate models agree with observations?, Philosophical Transactions of the Royal Society of London A: Mathematical, Physical and Engineering Sciences, 373, https://doi.org/10.1098/rsta.2014.0164, 2015.

Roach, L. A., Dörr J., Holmes, C. R., Massonnet, F., Blockley, E. W., Notz, D., Rackow, T., Raphael, M. N., O'Farrell, S. P., Bailey, D. A., and Bitz, C. M. (2020). Antarctic sea ice area in CMIP6. Geophysical Research Letters, 47, e2019GL086729. https://doi.org/10.1029/2019GL086729.

---

## Author Comment (AC2)

Reviewer #2: The authors address the topic of parameter sensitivity for simulating Antarctic sea ice, an important topic, and demonstrate some parameters which are particularly important. This is a valuable piece of work that is worthy of publication.

However, I must disagree with reviewer #1 that the paper was well-written. I may not have the requisite expertise to fairly review the paper, but I found the manuscript to be incredibly hard to follow, the Methodology was poorly explained and in particular I was not sure how to interpret many of the figures. I've attached an annotated PDF but here are some general comments that I think the authors should address:

Thank you for your constructive comments. We have revised the manuscript by improving its clarity and readability. Please see below our response (blue text) to each specific comment point by point.

1. Methods - I'm not very cognizant of Emulator techniques, and I struggled to understand many of the resulst because I wasn't sure how to interpret most of the metrics. I think some concise and clear description of what the metrics indicate, why different emulators are employed etc would be helpful

We have reformulated the methods section to make it clearer and more concise.

- In section 2.2 we rephrase the process in Fig. 1, clarifying more clearly what each metric means and why the GP emulator should be used. The methods in each section below correspond to one of the steps in Fig. 1.

- Section 2.3 was separated from section 2.2 of the original manuscript to specifically describe the method of sampling the 18 model parameters.

- Section 2.4 has been made smoother and more concise to highlight that the most critical step is the selection of the covariance function in the GPy software, and the sentence in the original manuscript about how the software is calculated has been removed to avoid diverting the reader's attention.

- The notation in section 2.5 on the first term on the left in Eq. (6) has been represented more specifically, which now reads (Line 176): "where the left-handside term is the change or dadt (also referred to specifically as dC/dt and dV/dt for changes in SIC and SIV respectively)". And these two notations are unified throughout the text and the figures.

2. Figures. There are a lot of figures, many of which are buried in the Supplemental Material. I think where an entire paragraph or more is devoted to a figure, it cannot be fairly described as Supplemental. There seemed to be no logical reason why some figures are in the ms and some in the Supplement. I don't think the Authors have given sufficient thought on what their key messages are, and what figures are required to convey that message (and more importantly which are not). I suspect that the ms will get much more impact with fewer but well-explained figures, appropriately placed in the main text.

We have moved Figs. B4 and B5 from the original manuscript to the main text (now Figs. 4 and 6) as suggested by the reviewer, given that the text uses two paragraphs to describe the two figures. In addition, we have split section 3.2 into two subsections (3.2.1 and 3.2.2) and have started section 3.2 with a paragraph that briefly summarises the results that this section seeks to present. The headings in section 3.2.1 (Model ensemble mean and standard deviation) summarise exactly the information expressed in Figs. 3 to 6. Subsequent to the above changes, the reasons for placing the figures in the main text and in the appendix respectively should now no longer confuse the reader.

At this point, we believe that the main text contains all the key information we have intended to convey. However, precisely because of this, it seems impossible to put fewer figures in the text, as the removal of any one figure would make the study incomplete.

3. Parameter names; the authors consistently refer to the model variable name(or symbol as per Table 1 header) for each parameter (e.g. Cd_ice) rather than a more physicallymeaningful long name. For anyone not intimately familiar with SI3 this makes it really hard to follow, without frequently referring back to Table 1. Since the authors focus on only 4-5 parameters, perhaps be a little kinder to the reader and use long names ('Description' in the Table 1 header) in the text.

All parameter names appearing in the main text have been replaced with long names, especially the key ones, and we have additionally appended symbols to the long names to make it easier for the reader to follow and correspond to the figures. For example,

- Lines 383-385: … Fig. 9 shows that in summer the $RMSE_{SICB}$ may also be sensitive to the minimum floe diameter for lateral melting parameter (rn_dmin) and the magnitude of the damping on salinity (rn_deds) …

- Lines 434-435: Consequently, in contrast to the reduction of snow thermal conductivity (rn_cnd_s) which inhibits ice freezing …

- Lines 456-458: With the exception of snow thermal conductivity (rn_cnd_s), ice-ocean drag coefficient (rn_cio) and air-ice drag coefficient (Cd_ice), whose physical effects have been elucidated, the adv term in the winter SIV budget is also sensitive to ice strength (rn_pstar) (Fig. 12a).

4. Sub-sections. It took me a while to work out the aim of some of the subsections. Some slightly more meaningful sub-headings would be helpful, and maybe an opening sentence (e.g. in the last section we showed which parameters the model is most sensitive to, we now explore the optimal values of those parameters to match the observed budget..)

Thanks for the good idea to add some link-up sentences at the beginning of each section. We checked heading of each sub-section and added the opening sentence accordingly to improve the readability of the entire manuscript, e.g.:

- Section 3.2 (Lines 242-247): The diagnostics of the SIC and sea ice thickness of the model ensemble in the last section show that .... In this section we first calculated the SIC budget and SIV budget for the ensemble of 449 model runs by applying the same approach as for the calculation of the observed SIC budget (cf. Fig. B1), and then ...

- Section 3.3 (Lines 333-336): Based on the results of the last section, the area integrals of adv and res in the SIC (and SIV) budget and the $RMSE_{SICB}$ are used as the metrics to assess the sensitivity of the model's sea ice budget to 18 parameters in this section …

- Section 4.1 (Lines 416-418): Several parameters have been identified in Sections 3.3 and 3.4 as having a significant impact on the simulated SIC and SIV budgets in the Southern Ocean. In this section we present how these parameters specifically act on the SIC and SIV budget by looking at the impact of parameter changes on the cumulative distribution function (CDF) in the PAWN method.

- Section 4.3 (Lines 502-504): The previous sections have shown the sensitivity of the simulated sea ice budget to parameters and there are a number of parameter sets that are recommended (Table 3), in this section we provide further insight into how these parameter sets perform in terms of other metrics.

Please also note the supplement to this comment:

https://gmd.copernicus.org/preprints/gmd-2022-170/gmd-2022-170-RC2-supplement.pdf

Line 18: "sensitivity" change to "sensitive"

Revised.

Line 20: change "better quality of" to "optimised"

Revised.

Line 37: remove "Whereas"

Removed.

Line 56: "at time", something missing here?

Yes, we have modified it to "at a time".

Line 70: the switch from writing about emulators to the purpose of this paper is quite abrupt. I would suggest adding a sentence that clarifies the relevance of the text on emulators to this paper

The new text reads (Line 73): "In this paper, our research objective is to quantify the

sensitivity … in a coupled ocean-sea ice model by constructing a GP emulator, …" to clarify the relationship between the emulator and our research objectives: the emulator is our key method for achieving this sensitivity experiment.

Line 101: "the other frequencies are" suggest changing to "...and 3 hours for all other surface boundary conditions."

Done.

Line 102: maybe add comment on whether this includes Antarctic ice mass loss (which of course is most relevant for Antarctic sea ice)

The added text reads (Line 102): "The continental discharge rates followed the climatological dataset of Dai and Trenberth (2002) and do not include ice mass loss in Antarctica."

Line 106: This sentence needs some work, the phrasing is a little odd and I don't quite understand what is being stated.

This sentence has now been reorganised as (Lines 108-110) "To investigate the sensitivity of sea ice budgets, we selected 18 parameters and determined their uncertainties (Table 1), which cover a number of important processes in sea ice modelling, such as ice/snow physical properties, ocean mixing and eddies, and ice-ocean/air-ice interactions." to make it more fluent and easier to follow.

Line 109: "elicited" change to "selected"?

Done.

Line 116: I struggle with this section, it needs a bit of editing to make it more generally comprehensible

This section has been largely reformulated. We tried our best to make it clear.

Line 200: comment on underestimation of divergence?

We agree that an overall overestimation of sea ice velocity observations by 5% may lead to a relative underestimation of divergence in the model results, however, we do not have sufficient evidence to suggest that this relationship is robust. For instance, even when calculated using NSIDC sea ice drift data (Tschudi et al. 2019), which has been reported to be underestimated, the model results are still underestimated compared to it (Nie et al., 2022), and the reason for the model underestimation of divergence is still under investigation.

Line 246: Spatial differencing always produces a 'noisy' output, even using the 3x3 grid cell smoothing that the authors applied to the velocity data. I would be hesitant to pay much heed to relatively small scale features such as this, which are just as likely to be artifacts of the observations

Thanks for your thoughts, we agree with the reviewer that the satellite-derived observation would contain some noise, even with the 3x3 grid filter. However, we are not sure if it would be good to consider this part of the convergence to be noise and accordingly unimportant. Because a) the observed convergence and divergence in the marginal ice region are almost interleaved, i.e. they are comparable in scales, rather than the convergence having a smaller scale (Fig. B1); b) as a comparison, there is also significant sea ice convergence in marginal sea ice regions even with a much smoother 7x7 grid filter (Holland and Kimura, 2016; their Fig. 3). Therefore, we would prefer to keep the original text.

Line 253: There's a whole paragraph devoted to this figure, so why is it stuffed into the appendices?

We agree that moving Fig. B4 from the appendix to the text would be a better choice, as we have already devoted a whole paragraph to the figure. We have now modified it and also moved Fig. B5 from the Supplementary Material into the main text.

Line 264: I can't help but think that this should be in the appendices and fig B4 - which shows the sensitivity of these terms to parameter changes - in the main text

We have followed the reviewer's suggestion to move Fig. B4 to the text, and to change the title of section 3.2.1 to "Model ensemble mean and standard deviation" to better summarise the information in Figs. 3 to 6.

Line 278: should really change the figure captions to dV/dt, not dadt, ideally change fig 3 to dC/dt (dadt is a bit sloppy)

Thanks for your suggestion, we have replaced "dadt" with "dC/dt" or "dV/dt" when it has a specific indication, both in the main text and in all the related figures, e.g.:

- Line 176: where the left-hand-side term is the change or dadt (also referred to specifically as dC/dt and dV/dt for changes in SIC and SIV respectively).
- Lines 249-251: As can be seen in Fig. 3, the spatial pattern characteristics of the ensemble mean of dC/dt and adv for each season are generally consistent with observations. The magnitudes of the model ensembles of dC/dt and adv are significantly larger due to the fact that the observed ice drift has some missing values and the dC/dt term …
- Figs. 3 to 7.

Line 285: Note that since SIV is a conserved term (unlike SIC), the 'residual' can be correctly called the 'thermodynamic' contribution.

Thank you for the clarification, we have now added a sentence to clarify it (Lines 295-297): "The residual term, which equals the thermodynamic contribution as SIV is conserved, still has the largest standard deviation as it retains the deviations of the other terms."

Line 287: insert '...and time... ' after 'area'

Added.

Line 291: numerically, the dynamic terms of SIV (adv + div) should be exactly zero when integrated over the Southern Ocean. (not necessarily for SIC, since it's not a conserved quantity)

We agree, and we believe that the possible reason for the sum of the area integrals of adv and div to be zero for SIC is that in the Southern Ocean, where sea ice is close to free drift, the non-conservation due to sea ice ridging/rafting (r in Eq. 5) is negligible relative to the dominant role of thermodynamics (f in Eq. 5), as revealed by Uotila et al. (2014) and Holland and Kimura (2016).

We have reformulated this sentence and it now reads (Lines 305-308) "For SIV this is because these two processes do not change the total amount of sea ice, and for SIC this also holds approximately, considering that in the Southern Ocean sea ice is close to free drifting and the non-conserve nature of SIC due to ridging can be neglected (Uotila et al., 2014; Holland and Kimura, 2016)."

Line 292: This isn't correct. It's true that the Southern Ocean integral of the the dynamic SIV term should numerically be zero. BUT those terms still effect the thermodynamic term. Consider a completely static icepack with no velocity at all - it will have a much lower thermodynamic freeze/melt rate because divergence/advection will not maintain areas of open ocean (e.g. polynyas). Numerically though the net dynamic term is the same as a realistic representation, i.e. zero

Thanks for the correction and the reasonable example. Since numerically the spatial integrals of adv and div do cancel each other out (and thus the integrals of dadt and res are almost identical), the sensitivity analysis only for res and adv does not affect the conclusions of our study. We have modified this sentence to make it flow more logically with the preceding text (Lines 308-309), "Therefore, when studying the effects of model parameter uncertainty on sea ice budgets in the following sections, it is only necessary to only use the area integrals of res (or dadt) and adv (or div)."

Line 305: insert 'net' after 'of'
Revised.

Line 329: how do these thermodynamic parameters have an influence on the dynamic term?

The thermodynamic and dynamic processes are interdependent, one example would be a case given by Uotila et al. (2014) "where the ice melt decreases the sea-ice concentration and thickness, and consequently results in a faster moving sea ice, which in turn affects the divergence and advection."

Line 331: 'PWAN' change to 'PAWN'

Fixed.

Line 338: Although you've already defined the term in line 330, it's really hard to remember what all the parameter names represent, so I would advice using the long names each time as well

We are now using long names throughout the text and have added a short-hand symbol at the end of the long name to enhance readability.

Line 353: "perfectly" change to "well"

Revised.

References:

Holland, P. R. and Kimura, N.: Observed concentration budgets of Arctic and Antarctic sea ice, J. Clim., 29(14), 5241–5249, doi:10.1175/JCLI-D-16-0121.1, 2016.

Holmes, C. R., Holland, P. R. and Bracegirdle, T. J.: Compensating Biases and a Noteworthy Success in the CMIP5 Representation of Antarctic Sea Ice Processes, Geophys. Res. Lett., 46(8), 4299–4307, doi:10.1029/2018GL081796, 2019.

Nie, Y., Uotila, P., Cheng, B., Massonnet, F., Kimura, N., Cipollone, A., and Lv, X.: Southern Ocean sea ice concentration budgets of five ocean-sea ice reanalyses, Clim. Dyn., https://doi.org/10.1007/s00382-022-06260-x, 2022.

Tschudi, M. A., Meier, W. N., and Scott Stewart, J.: An enhancement to sea ice motion and age products at the National Snow and Ice Data Center (NSIDC), Cryosphere, 14, 1519–1536, https://doi.org/10.5194/tc-14-1519-2020, 2020.

Uotila, P., Holland, P. R., Vihma, T., Marsland, S. J. and Kimura, N.: Is realistic Antarctic sea-ice extent in climate models the result of excessive ice drift?, Ocean Model., 79, 33–42, doi:10.1016/j.ocemod.2014.04.004, 2014.

---

## Author Response (AR2)

Reviewer #1:

The revised version of this manuscript addresses the primary concerns of my first review. The addition of long names in the text for the primary parameters is particularly helpful and makes this much easier on the reader. I also find the colors in Figure 10 much improved as well.

Thank you for the constructive comments. We have revised manuscript accordingly. Please see below our response (blue text) to each specific comment point by point.

I have a few relatively minor suggestions.

I believe that there may be a couple different "audiences" for this paper: model developers (particularly sea ice in within the context of global climate models), and climate model users. The extensive details and figures along with the results will serve the former community particularly well. I also believe the paper will be of benefit to the latter group as well (of which I consider myself a member) and to that aim I suggest adding to the abstract two important conclusions (in my opinion) that are currently found in the "recommended parameters" and "conclusion" sections are worth highlighting up front for the benefit of the larger modeling community:

1. "key to reproducing a realistic SIC budget of an ice-ocean coupled model driven by atmospheric analysis is to simulate realistic sea ice velocities" (lines 565-566)

2. "accurate modelling of the SIC budget does not appear to be possible by simply changing the atmospheric forcing product or tuning ocean model's parameters, as the atmospheric forcing itself is systematically biased" (lines 514-518)

These are really important points to bear in mind when using climate models to explore polar region dynamics, mechanisms, change, etc.

We agree with the reviewer that including these two conclusions in the abstract would help the audience for this paper. We have now added the following sentence to Line 21 to highlight these two points: "This implies that a more accurate calculation of ice

velocity is the key to optimising the SIC budget simulation, which is unlikely to be achieved perfectly by simply tuning the model parameters in the presence of biased atmospheric forcing."

And the last sentence in the abstract has also been changed to "Nevertheless, ten combinations of NEMO4.0-SI$^3$ model parameters were recommended as they could yield better sea ice extent and SIC budgets than using the standard values." to make the text read more smoothly.

I suggest adding 1-2 sentences summarizing your paragraph that response to my question about how these parameter set recommendations might change optimal values in a warming world (to paragraph that ends at line 159). I found the response in "response to reviewers" quite helpful, for example, in that I didn't imagine how air-ice drag coefficient might change due to changing surface roughness and atmospheric stability in a warming world. Essentially what I would like in the paper, if possible, is a couple sentences summarizing how different conditions may result in different optimizations and whether or not the impacts would be "large". I don't expect detailed answer here to this complicated question – clearly beyond the scope of this work – just something to help what I consider "climate model users" (not sea ice model developers per se) understand how or which parameters might change and how large of change – i.e. should I still be able to trust sea ice component of a climate model that has parameters optimized from current conditions to help me understand future polar climate change? Or not? Or what should I bear in mind when using sea ice model in somewhat different climate states? Or even how or why they might change but no idea exactly how much or if it will be of impact on a larger scale? Just a summary sentence or two that would help the reader in the same way your response to my review helped me…

Thanks for the suggestion, and we agree that add some discussion on this topic would be helpful to climate model users. The Line 159 correspondings to Gaussian process emulator formulation, which we do not think is where the reviewer was referring to.

We have now added one sentence at the end of the paper (Lines 573-575):
"The recommended parameter sets are determined based on the current climate scenario, and their optimal values are expected to change to some extent when applied to simulate sea ice in a warming world. In general, one might expect the global or hemispheric optimal parameter values to change little because even now global sea-ice models can reasonably reproduce regional sea ice characteristics, ideally associated with a wide range of optimal parameter values."

In addition to the above general comments I have some specific minor suggested changes outlined below. I also recommend if possible to have a good editor go through the manuscript for editorial changes – I've found some myself but am not skilled per se in this arena and have most likely missed some others.

Line

30 two Notz articles referenced are not in reference list
Thanks for your reminder, both references have now been added, and we have checked the reference specification again.

87 add "thick" (m thick) as a little confusing as written
Revised.

90 add "scheme"
Done.

135 replace "divide" with "divides"
Corrected.

147 replace "practivally" with "practically"
Corrected.

203 replace "with" with "while"

Done.

217 replace "observation" with "observations"

Corrected.

235 "detached" ? huh? Tails of distribution larger in winter than summer?

We have revised this sentence to "Additionally, the SIV cycles show a larger spread in winter than in summer, which is opposite to that of SIE cycles" to make it clearer.

253, 254, 258 replace "observation" with "observations"

Thank you for the correction. We have used several SIC observations to verify the SIE/SIA results, so changing "observation" to "observations" in line 217 is necessary. However, only one observational SIC budget product (i.e., calculated by CDR SIC and KIMURA ice drift) was used to diagnose the modelled SIC budget results in Section 3.2, so we replaced "observation" with "observational data" in these lines. Other places where "observation" was used have also been changed to "observational data" or "observations" accordingly (e.g., Line 188, 285, 383, 405).

281 "very little different"…..rephrase

We have rephrased this sentence to "The spatial pattern of the divergence of SIV does not differ much from that of SIC".

303. delete "a" in "implement a method"

Revised.

389. rn_beta and rn_dmin – suggest helping reader with long names…

We have added their long names afterwards to help readers.

458 "weaker ice is more easily to deform and increase ice thickness" ? doesn't make sense

Docquier et al. (2017) carried out a detailed study of the relationship between Arctic sea ice drift and ice strength modelled by NEMO-LIM3.6, and shows that "higher values of P* generally lead to lower sea ice deformation and lower sea ice thickness", and the results of our sensitivity analysis (Fig. 12a) support that this conclusion also holds in the Antarctic. We have now cited this paper to make the statement in the text more convincing (Line 462).

469-470 "always shows a similar high sensitivity to …and…. " to "show similarly high sensitivities to …"
Revised.

483 delete "question"
Done.

Reference:

Docquier, D., Massonnet, F., Barthélemy, A., Tandon, N. F., Lecomte, O., and Fichefet, T.: Relationships between Arctic sea ice drift and strength modelled by NEMO-LIM3.6, Cryosphere, 11, 2829–2846, https://doi.org/10.5194/tc-11-2829-2017, 2017.